# AN EXPLORATION OF NON-EUCLIDEAN GRADIENT DESCENT: Muon AND ITS MANY VARIANTS

## ABSTRACT

To define a steepest descent method over a neural network, we need to choose a norm for each layer, a way to aggregate these norms across layers, and whether to use normalization. We systematically explore different alternatives for aggregating norms across layers, both formalizing existing combinations of Adam and the recently proposed Muon as a type of non-Euclidean gradient descent, and deriving new variants of the Muon optimizer. Through a comprehensive experimental evaluation of the optimizers within our framework, we find that Muon is sensitive to the choice of learning rate, whereas a new variant we call MuonMax is significantly more robust. We then show how to combine any Non-Euclidean gradient method with model based momentum (known as Momo). The new Momo variants of Muon are significantly more robust to hyperparameter tuning, and often achieve a better validation score. Thus for new tasks, where the optimal hyperparameters are not known, we advocate for using Momo in combination with MuonMax to save on costly hyperparameter tuning.

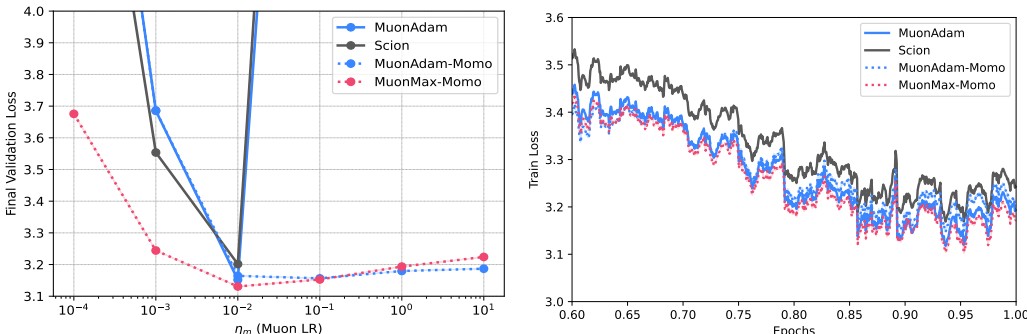

Figure 1: Learning rate sweep for training GPT2-Large (774M params) on SlimPajama with 1B tokens. **Left:** Final validation loss for various learning rates. MuonAdam and Scion require precise tuning, whereas our MuonAdam-Momo and MuonMax-Momo achieve low loss for a significantly wider range of learning rates. **Right:** Training loss (with tuned LRs) for the last 40% of steps.

## 1 INTRODUCTION

The recently proposed Muon optimizer (Jordan et al., 2024b) has generated increasing interest due to its efficiency for training language models (Pethick et al., 2025; Liu et al., 2025). This algorithm was initially introduced (Bernstein & Newhouse, 2024a; Jordan et al., 2024b) and often interpreted (Pethick et al., 2025; Kovalev, 2025; Fan et al., 2025) as steepest descent with respect to the spectral norm for each weight matrix in a neural network.

However, this interpretation does not entirely apply for practical implementations of Muon. In practice, Muon is used side-by-side with another optimizer, where hidden weight matrices are trained with Muon, and all other parameters by Adam (Jordan et al., 2024b; Liu et al., 2025; Jordan et al., 2024a) or SignDescent (Pethick et al., 2025). We will refer to this combination as MuonAdam

throughout, see Algorithm 1 in the appendix. Furthermore, for the weight matrices only the normalized version of Muon has been explored in practice.

Here we aim to strengthen the theoretical foundation of MuonAdam and develop new optimizers by systematically investigating different design choices which have not been explored. We introduce a framework for steepest descent on the entire space of network parameters, which involves a choice of norm for each layer, a *product norm* to aggregate norms across layers, and whether to normalize updates. This framework encompasses MuonAdam and other variations, which provides a more principled interpretation of these algorithms as genuine steepest descent on the entire space of network parameters, and also opens a design space for previously unexplored Muon-type algorithms.

One such unexplored variant is what we call MuonMax, that arises from a new product norm and does not use update normalization. The updates of MuonMax depend on the nuclear norm of the momentum from every weight matrix, which is slightly less efficient per-step than Muon. To make MuonMax more efficient, we introduce a stale approximation of these nuclear norms, which can be implemented with near-identical memory and 5% additional wall-clock time per step as Muon.

Now that we can frame MuonAdam and other variants as a type of steepest descent, we can import other tools used for steepest descent gradient methods. One such tool is Momo (Schaipp et al., 2024), an adaptive step size based on model truncation (Asi & Duchi, 2019b) that increases robustness to learning rate tuning. We extend the Momo step size to general steepest descent for arbitrary norms and subsequently apply it to the algorithms in our framework.

We perform a systematic evaluation of many algorithms in our framework for training GPT models with up to 774M parameters for language modeling on the FineWeb (Penedo et al., 2024) and SlimPajama (Soboleva et al., 2023) datasets with up to 6B tokens. We find that MuonMax-Momo consistently matches or outperforms MuonAdam and Scion (Pethick et al., 2025) while enjoying a much larger range of competitive learning rates, meaning that MuonMax-Momo is much less sensitive to tuning. We also observe that Momo increases tuning robustness for all variations and that our stale nuclear norm approximation causes negligible change in performance, while decreasing wall-clock time per iteration. Our contributions are:

1. **Formalizing** MuonAdam. We introduce a steepest descent framework that encompasses the practical implementation of Muon (with Adam used for a subset of parameters), demonstrating that even these side-by-side optimizers can be interpreted as steepest descent with respect to certain norms on the space of *all* network parameters. This solidifies the theoretical foundation for practical variants of Muon, and sheds light on unexplored aspects of Muon's design. Our framework also includes Scion and other existing variants.

2. **Defining non-Euclidean** Momo. We show how to incorporate the adaptive step size Momo with any steepest descent algorithm, which we find significantly increases robustness to the learning rate tuning.

3. MuonMax: **New practical, robust variant of** Muon. We propose a new optimizer, MuonMax, which arises within our framework from a novel product norm. With a stale approximation of the nuclear norm of each layer's momentum, MuonMax has near-identical memory cost and 5% additional wall-clock time per iteration compared to Muon.

4. **Systematic Evaluation.** We perform a comprehensive evaluation of optimizers in our framework for language modeling. MuonMax-Momo consistently matches or outperforms Muon and other baselines while widening the range of competitive step size choices by several orders of magnitude.

**Notation.** We use $\langle \cdot, \cdot \rangle$ to denote the Euclidean inner product on $\mathbb{R}^d$ or on products of the form $\mathbb{R}^{d_1} \times \ldots \times \mathbb{R}^{d_n}$ naturally by viewing elements of $\mathbb{R}^{d_1} \times \ldots \times \mathbb{R}^{d_n}$ as elements of $\mathbb{R}^{d_1 + \ldots + d_n}$. Note that for matrices, which can also be viewed as elements of $\mathbb{R}^{mn}$, this inner product is consistent with the trace inner product, since $\text{Tr}(\boldsymbol{A}^T \boldsymbol{B}) = \langle \text{vec}(\boldsymbol{A}), \text{vec}(\boldsymbol{B}) \rangle$.

## 2 RELATED WORK

**Muon.** The use of *Spectral descent*, that is steepest descent with respect to the spectral norm, on deep neural networks dates back to Carlson et al. (2015a;b). Muon is the combination of spectral

descent with momentum (Bernstein & Newhouse, 2024a), and a carefully crafted polynomial algorithm for computing the polar factor (Jordan et al., 2024b). Recent work has designed an optimal such polynomial algorithm for the polar factor called PolarExpress (Amsel et al., 2025), which we use in our Muon implementation. Pethick et al. (2025) introduced Scion, which uses SignSGD with momentum (instead of Adam) to train non-matrix parameters. Liu et al. (2025) scaled Muon to train a 16B parameter language model with 5.7T tokens. Several works have developed theory of Muon's convergence (Li & Hong, 2025; Kovalev, 2025; Riabinin et al., 2025) and implicit bias (Tsilivis et al., 2025; Fan et al., 2025).

Most similar to ours is the line of work developing the modular norm (Bernstein & Newhouse, 2024a; Large et al., 2024; Bernstein & Newhouse, 2024b). This line of work also argues that we should perform steepest descent on the entire space of network parameters, instead of separately at each layer, and focuses on steepest descent with respect to a particular norm called the modular norm. This norm enables Lipschitz continuity of the neural network with respect to both weights and inputs. In this work, we take an orthogonal approach, where we develop a general theory of steepest descent on product spaces, and numerically investigate many possible norms on these spaces. We are not aware of any existing evaluation of steepest descent with respect to the modular norm.

**Model Truncation.** Gradient descent can be viewed as using the local linearization of the loss as a *model* of the loss itself. If we know a lower bound of the loss, for instance most loss functions are positive, then we can improve this linear model by *truncating* the model at this lower bound (Asi & Duchi, 2019a). Follow-up work emphasizes the importance of such model choices in stochastic optimization (Asi & Duchi, 2019b), and extends the framework to minibatch settings (Asi et al., 2020). Using model truncation often leads to methods that are more stable and easier to tune (Loizou et al., 2021; Davis & Drusvyatskiy, 2019; Meng & Gower, 2023; Schaipp et al., 2023). Recently Schaipp et al. (2024) showed how to combine momentum with model truncation. Furthermore, Chen et al. (2022) combine parameter-free coin betting methods with truncated models.

## 3 STEEPEST DESCENT ON NEURAL NETWORKS

Let $F : \mathbb{R}^d \to \mathbb{R}$ be the loss function, and $\|\cdot\|$ be any norm on $\mathbb{R}^d$. We define the *Linear Minimization Oracle* (LMO) and the *dual norm* as

$$\mathsf{LMO}_{\|\cdot\|}(\boldsymbol{v}) = \operatorname*{arg\,min}_{\|\boldsymbol{u}\|=1} \langle \boldsymbol{u}, \boldsymbol{v} \rangle, \quad \text{and} \quad \|\boldsymbol{v}\|_* = \max_{\|\boldsymbol{u}\|=1} \langle \boldsymbol{u}, \boldsymbol{v} \rangle, \tag{1}$$

respectively. When the norm is clear from context, we will omit the subscript and write LMO. Throughout we denote the stochastic gradient at step $t$ by $\boldsymbol{g}_t$, and the momentum buffer $\boldsymbol{m}_t$ which is an exponential moving average of stochastic gradients, i.e. $\boldsymbol{m}_t = \beta \boldsymbol{m}_{t-1} + (1-\beta)\boldsymbol{g}_t$ for $\beta \in [0, 1)$.

### 3.1 CONSTRAINED VS REGULARIZED STEEPEST DESCENT

For a single weight matrix, the Muon update is often motivated as the LMO (Pethick et al., 2025) with respect to the spectral norm. The following proposition shows that for a general norm, updating in the direction of LMO($\boldsymbol{m}_t$) is equivalent to minimizing a first-order Taylor approximation of $F$ around $\boldsymbol{w}_t$, with a constraint on the update's norm and approximating $\nabla F(\boldsymbol{w}_t) \approx \boldsymbol{m}_t$.

**Proposition 3.1.** [Constrained Steepest Descent] The CSD update is given by

$$\boldsymbol{w}_{t+1} = \operatorname*{arg\,min}_{\|\boldsymbol{w}-\boldsymbol{w}_t\| \leq \eta} \{F(\boldsymbol{w}_t) + \langle \boldsymbol{m}_t, \boldsymbol{w} - \boldsymbol{w}_t \rangle\} = \boldsymbol{w}_t + \eta\, \mathsf{LMO}(\boldsymbol{m}_t). \tag{2}$$

The ball constraint above ensures that we only use the Taylor approximation close to its center $\boldsymbol{w}_t$, but another natural choice is to use regularization instead of a constraint as follows.

**Proposition 3.2.** [Regularized Steepest Descent] The RSD update is given by

$$\boldsymbol{w}_{t+1} = \operatorname*{arg\,min}_{\boldsymbol{w}} \left\{F(\boldsymbol{w}_t) + \langle \boldsymbol{m}_t, \boldsymbol{w} - \boldsymbol{w}_t \rangle + \tfrac{1}{2\eta} \|\boldsymbol{w} - \boldsymbol{w}_t\|^2\right\} = \boldsymbol{w}_t + \eta\|\boldsymbol{m}_t\|_* \mathsf{LMO}(\boldsymbol{m}_t) \tag{3}$$

In the case without momentum (i.e. $\beta = 0$), both of these algorithms have appeared throughout the literature under the name steepest descent, but the recent line of work around MUON (Jordan et al., 2024b; Bernstein & Newhouse, 2024b; Pethick et al., 2025; Liu et al., 2025) has mostly focused on the constrained variant. To the best of our knowledge, the only work which considered the regularized variant over the space of all parameters was Bernstein & Newhouse (2024a). Lau et al. (2025) also use the regularized interpretation of MUON on a per layer basis instead of the entire product space.

Notice that CSD and RSD have the same update direction, but with regularization the update magnitude is multiplied by the dual norm of the momentum. Therefore, the primal norm of the update $\|\boldsymbol{w}_{t+1} - \boldsymbol{w}_t\|$ is $\eta$ for CSD and $\eta\|\boldsymbol{m}_t\|_*$ for RSD. Intuitively, CSD enforces a *normalized update*.

## 3.2 PRODUCT NORMS

To describe steepest descent, we first need a norm over the space of *all* network parameters (Bernstein & Newhouse, 2024a). Instead of flattening all parameters into a single vector, we consider the Cartesian product $\boldsymbol{W} = (\boldsymbol{w}^1, \boldsymbol{w}^2, \ldots, \boldsymbol{w}^n)$ of network parameters (where each $\boldsymbol{w}^i$ could be a flattened weight matrix, a bias vector, etc). We assign a norm $\|\cdot\|_{(i)}$ for parameter $\boldsymbol{w}^i$, then aggregate these norms into a single norm on the product space. Two natural examples of product norms are

$$\|\boldsymbol{W}\|_\infty := \max_{1 \leq i \leq n} \|\boldsymbol{w}^i\|_{(i)}, \quad \text{and} \quad \|\boldsymbol{W}\|_2 := \sqrt{\sum_{i=1}^n \|\boldsymbol{w}^i\|_{(i)}^2}. \tag{4}$$

Computing the steepest descent direction with respect to a product norm requires: the linear minimization oracle (LMO) and the dual norm of the product norm. As we show next, both can be expressed in terms of the underlying per-parameter norms and the norm used to aggregate them.

**Lemma 3.3.** [LMO and Dual of Product Norms] For each $i \in [n]$, let $g_i$ be a norm on $\mathbb{R}^{d_i}$, and let $f$ be a norm on $\mathbb{R}^n$, and denote their dual norms as $g_{i,*}$ and $f_*$, respectively. Then the product norm $h : \mathbb{R}^{d_1} \times \ldots \times \mathbb{R}^{d_n} \to \mathbb{R}$ defined by

$$h(\boldsymbol{w}^1, \ldots, \boldsymbol{w}^n) = f\big(g_1(\boldsymbol{w}^1), \ldots, g_n(\boldsymbol{w}^n)\big) \tag{5}$$

is indeed a norm, and its LMO and dual norm are given by

$$\mathsf{LMO}_h(\boldsymbol{w}^1, \ldots, \boldsymbol{w}^n) = (\phi_1\mathsf{LMO}_{g_1}(\boldsymbol{w}^1), \ldots, \phi_n\mathsf{LMO}_{g_n}(\boldsymbol{w}^n)) \tag{6}$$

$$h_*(\boldsymbol{w}^1, \ldots, \boldsymbol{w}^n) = f_*(g_{1,*}(\boldsymbol{w}^1), \ldots, g_{n,*}(\boldsymbol{w}^n)), \tag{7}$$

where $(\phi_1, \ldots, \phi_n) := -\mathsf{LMO}_f(g_{1,*}(\boldsymbol{w}^1), \ldots, g_{n,*}(\boldsymbol{w}^n))$.

We can now compute steepest descent updates (both constrained and regularized) with respect to the product norms $\|\cdot\|_\infty$, $\|\cdot\|_2$, or any other product norm by plugging the LMO and dual of each product norm into the steepest descent definitions (Equation 2 and Equation 3).

Denoting by $\boldsymbol{m}_t^i$ the momentum buffer of parameter $i$, the updates for each parameter $\boldsymbol{w}^i$ are:

$$\text{CSD w.r.t. } \|\cdot\|_\infty: \quad \boldsymbol{w}_{t+1}^i = \boldsymbol{w}_t^i + \eta\, \mathsf{LMO}_{\|\cdot\|_{(i)}}(\boldsymbol{m}_t^i) \tag{8}$$

$$\text{RSD w.r.t. } \|\cdot\|_\infty: \quad \boldsymbol{w}_{t+1}^i = \boldsymbol{w}_t^i + \eta\Big(\sum_{j=1}^n \|\boldsymbol{m}_t^j\|_{(j),*}\Big)\mathsf{LMO}_{\|\cdot\|_{(i)}}(\boldsymbol{m}_t^i) \tag{9}$$

$$\text{CSD w.r.t. } \|\cdot\|_2: \quad \boldsymbol{w}_{t+1}^i = \boldsymbol{w}_t^i + \eta\frac{\|\boldsymbol{m}_t^i\|_{(i),*}}{\sqrt{\sum_{j=1}^n \|\boldsymbol{m}_t^j\|_{(j),*}^2}}\mathsf{LMO}_{\|\cdot\|_{(i)}}(\boldsymbol{m}_t^i) \tag{10}$$

$$\text{RSD w.r.t. } \|\cdot\|_2: \quad \boldsymbol{w}_{t+1}^i = \boldsymbol{w}_t^i + \eta\,\|\boldsymbol{m}_t^i\|_{(i),*}\mathsf{LMO}_{\|\cdot\|_{(i)}}(\boldsymbol{m}_t^i) \tag{11}$$

For the methods above, the update direction for each parameter $\boldsymbol{w}_t^i$ is always the LMO of $\boldsymbol{m}_t^i$, regardless of the choice of product norm. However, the magnitude $\phi_i$ of each parameter's update is determined by the product norm and the dual norms of each parameter's momentum. Therefore, different choices of the product norm amount to different *parameter-wise learning rates*.

## 3.3 INCORPORATING ADAM

Now we show how to represent the hybrid $\mathrm{MuonAdam}$ method as a steepest descent method. For parameters $\boldsymbol{\theta}$, the $\mathrm{Adam}$ update, where all vector operations are element-wise[1], is given by

$$\boldsymbol{\theta}_{t+1} = \boldsymbol{\theta}_t - \eta \frac{\boldsymbol{m}_t}{\sqrt{\boldsymbol{v}_t}+\epsilon}, \quad \text{and} \quad \boldsymbol{v}_{t+1} = \beta_2 \boldsymbol{v}_t + (1-\beta_2)\boldsymbol{g}_t^2 \tag{12}$$

$\mathrm{Adam}$ can be interpreted as steepest descent in two different ways.

**Proposition 3.4.** The $t$-th update of $\mathrm{Adam}$ is the CSD with step size $\eta$ with respect to the norm:

$$\|\boldsymbol{\theta}\|_{\mathrm{ada}\infty} := \left\|\mathrm{Diag}\big(\tfrac{\sqrt{\boldsymbol{v}_t}+\epsilon}{|\boldsymbol{m}_t|}\big)\boldsymbol{\theta}\right\|_\infty \tag{13}$$

**Proposition 3.5.** The $t$-th update of $\mathrm{Adam}$ is the RSD with step size $\eta$ with respect to the norm:

$$\|\boldsymbol{\theta}\|_{\mathrm{ada}2} := \sqrt{\big\langle \mathrm{Diag}(\sqrt{\boldsymbol{v}_t}+\epsilon)\boldsymbol{\theta}, \boldsymbol{\theta}\big\rangle} = \left\|\mathrm{Diag}\big(\sqrt{\sqrt{\boldsymbol{v}_t}+\epsilon}\big)\boldsymbol{\theta}\right\|_2 \tag{14}$$

Thus $\mathrm{Adam}$ can be interpreted as either an adaptive trust-region sign descent (Balles & Hennig, 2018; Orvieto & Gower, 2025) or preconditioned gradient descent (Schaipp et al., 2024). A distinctive feature of these forms of steepest descent is that the norm changes over iterations.

## 3.4 THE WHOLE FRAMEWORK

For a given neural network, we partition the parameters as $\boldsymbol{W} = (\boldsymbol{W}^1, \ldots, \boldsymbol{W}^L, \boldsymbol{\theta})$, where $\boldsymbol{W}^1, \ldots, \boldsymbol{W}^L$ are the hidden weight matrices and $\boldsymbol{\theta}$ contains the remaining parameters flattened into a single vector. $\mathrm{MuonAdam}$ applies LMO updates w.r.t. the spectral norm for the hidden weight matrices, and uses $\mathrm{Adam}$ for the remainder of the parameters, with two separate learning rates for these side-by-side optimizers, shown in Algorithm 1 (Appendix A).

**Proposition 3.6.** $\mathrm{MuonAdam}$ (Algorithm 1) is exactly CSD with step size $\eta_m$ with respect to

$$\|\boldsymbol{W}\|_{\mathrm{muon}} = \max\left(\max_{\ell \in [L]} \|\boldsymbol{W}^\ell\|_{2\to 2}, \tfrac{\eta_m}{\eta_b}\|\boldsymbol{\theta}\|_{\mathrm{ada}\infty}\right). \tag{15}$$

The coefficient $\eta_m/\eta_b$ effectively allows for the use of different learning rates for hidden weight matrices compared to all other parameters; this is a crucial feature of $\mathrm{Muon}$'s speedrun implementation (Jordan et al., 2024b) and of other variations (Pethick et al., 2025; Liu et al., 2025).

Proposition 3.6 shows the precise sense in which $\mathrm{MuonAdam}$ is a steepest descent algorithm: it is constrained steepest descent with respect to a particular product norm that aggregates the spectral norm of each hidden weight matrix and an adaptive $\ell_\infty$ norm for all other parameters. This still leaves several other valid choices within our general steepest descent framework to explore: whether to use constrained or regularized steepest descent, which product norm to use ($\|\cdot\|_\infty$, $\|\cdot\|_2$), and which norm to assign to the non-matrix parameters ($\|\cdot\|_{\mathrm{ada}\infty}$, $\|\cdot\|_{\mathrm{ada}2}$, $\|\cdot\|_\infty$).

These three factors yield a design space for $\mathrm{Muon}$-type optimization algorithms, all of which are founded on the principle of steepest descent, and most of which are unexplored. Among these algorithms are several existing variations of $\mathrm{Muon}$, including $\mathrm{Scion}$ (Pethick et al., 2025) and $\mathrm{PolarGrad}$ (Lau et al., 2025) (see Appendix A.1 for the full statements).

**Stale dual norms.** Many of the updates we have presented so far require calculating dual norms of the momentum buffers (e.g. Equation 9 through Equation 11). If that norm is the spectral norm, this amounts to computing the nuclear norm of the momentum, which may appear costly, but actually the dual norm is easy to compute once we have computed the LMO, since $\|\boldsymbol{v}\|_* = \langle -\mathrm{LMO}(\boldsymbol{v}), \boldsymbol{v}\rangle$. However, in the case that updates are not separable across parameters, computing the dual norms of each momentum buffer in this way requires either additional memory (to store the layer-wise LMOs) or additional time (to compute the LMOs twice). To see why, consider the example of RSD with the $\|\cdot\|_\infty$ product norm (Equation 9), and assume for simplicity that all parameters are assigned the

---

[1]We omit the bias correction since this bias can be removed by correctly initializing the momentum buffers Schaipp et al. (2024). In any case it has little effect on performance (Orvieto & Gower, 2025).

spectral norm. For each layer $i$, the update $\boldsymbol{W}_{t+1}^i = \boldsymbol{W}_t^i - \eta \left( \sum_{j=1}^{L} \|\boldsymbol{M}_t^j\|_{\text{nuc}} \right) \text{polar}(\boldsymbol{M}_t^i)$ cannot be executed until $\|\boldsymbol{M}_t^j\|_{\text{nuc}} = \langle \text{polar}(\boldsymbol{M}_t^j), \boldsymbol{M}_t^j \rangle$ has been computed for every layer $j$. Crucially, the polar factors are used twice here: once to compute dual norms, and again to update weights. So, we can either store the polar factors for reuse (extra memory), or compute them twice (extra time); these options are sketched in the first two columns below.

| **Extra Memory** | **Extra Time** | **Stale Norms** |
|---|---|---|
| ```
d = 0
lmos = {}
for i in range(1, L+1):
    lmos[i] = -polar(M[i])
    d -= lmos[i].dot(M[i])

for i in range(1, L+1):
    W[i] += eta * d * lmos[i]
``` | ```
d = 0
for i in range(1, L+1):
    lmo = -polar(M[i])
    d -= lmo.dot(M[i])

for i in range(1, L+1):
    lmo = polar(M[i])
    W[i] += eta * d * lmo
``` | ```
new_d = 0
for i in range(1, L+1):
    lmo = -polar(M[i])
    new_d -= lmo.dot(M[i])
    W[i] += eta * old_d * lmo
old_d = new_d
``` |

The first option requires additional memory proportional to the size of the network, while the second option doubles the wall-clock time needed to compute polar factors. As an efficient approximation, we propose to reuse momentum dual norms from the previous step (shown in the third column), which can be implemented without storing or recomputing polar factors, and only requires a single additional scalar of memory for each layer. We found in our experiments that using these "stale" dual norms had near negligible effect on performance. Informally, we expect this approximation to work on the grounds that the momentum doesn't change too drastically in a single step, since

$$\boldsymbol{m}_t - \boldsymbol{m}_{t-1} = \beta \boldsymbol{m}_{t-1} + (1-\beta)\boldsymbol{g}_t - \boldsymbol{m}_{t-1} = (1-\beta)(\boldsymbol{g}_t - \boldsymbol{m}_{t-1}) \tag{16}$$

has small magnitude when $\beta$ is close to 1.

**A New Product Norm.** Our proposed algorithm $\text{MuonMax}$ is regularized steepest descent with respect to the following norm:

$$\|\boldsymbol{W}\|_{\text{MM}} := \sqrt{\left( \max_{\ell \in [L]} \|\boldsymbol{W}^\ell\|_{2 \to 2} \right)^2 + \frac{\eta_m}{\eta_b} \|\boldsymbol{\theta}\|_{\text{ada2}}^2} \tag{17}$$

This norm comes from assigning $\|\cdot\|_{\text{ada2}}$ to the non-matrix parameters, spectral norm to the matrix parameters, then aggregating both using the standard $\|\cdot\|_2$ Euclidean norm. We denote the corresponding product norm as $\|\cdot\|_{\text{hyb}}$, defined in Equation 127 of Appendix C.

## 4 Model Truncation

Beyond a more solid theoretical footing for Muon-type algorithms, our steepest descent framework also offers practical benefits: techniques designed for SGD (or normalized SGD) can now be easily adapted for Muon-type algorithms by generalizing to arbitrary norms. In this section, we generalize Momo (Schaipp et al., 2024) for steepest descent with respect to arbitrary norms.

Recall that both variations of steepest descent are motivated by locally minimizing a first-order Taylor approximation of the loss around the current iterate. Momo makes use of *model truncation* (Asi & Duchi, 2019b), which leverages knowledge of a loss lower bound $F_*$ to construct a better approximation of the loss which is more accurate than a Taylor approximation. In Momo, this model also incorporates information from the history of gradients and losses through momentum.

Denote $\rho_{i,t} = (1-\beta)\beta^{t-i}$, so that $\boldsymbol{m}_t = \sum_{i=0}^t \rho_{i,t} \boldsymbol{g}_i$, and denote by $F_t(\boldsymbol{w}_t)$ the minibatch loss at step $t$. Then for each $t$, we can build a model of the loss around $\boldsymbol{w}_t$ as a weighted average of first-order Taylor approximations centered at each iterate $\boldsymbol{w}_i$:

$$F(\boldsymbol{w}) \approx \sum_{i=0}^t \rho_{t,i} \left( F_i(\boldsymbol{w}_i) + \langle \boldsymbol{g}_i, \boldsymbol{w} - \boldsymbol{w}_i \rangle \right) \tag{18}$$

$$= \sum_{i=0}^t \rho_{t,i} \left( F_i(\boldsymbol{w}_i) + \langle \boldsymbol{g}_i, \boldsymbol{w}_t - \boldsymbol{w}_i \rangle \right) + \sum_{i=0}^t \rho_{t,i} \langle \boldsymbol{g}_i, \boldsymbol{w} - \boldsymbol{w}_t \rangle \tag{19}$$

$$= \tilde{F}_t + \langle \boldsymbol{m}_t, \boldsymbol{w} - \boldsymbol{w}_t \rangle, \tag{20}$$

where on the last line we denoted $\tilde{F}_t := \sum_{i=0}^t \rho_{t,i} \left( F_i(\boldsymbol{w}_i) + \langle \boldsymbol{g}_i, \boldsymbol{w}_t - \boldsymbol{w}_i \rangle \right)$. Since $F(\boldsymbol{w}) \geq F_*$ for all $\boldsymbol{w}$, we can improve our model by truncating, or clipping, it at $F_*$:

$$F(\boldsymbol{w}) \approx \max \left( \tilde{F}_t + \langle \boldsymbol{m}_t, \boldsymbol{w} - \boldsymbol{w}_t \rangle, F_* \right).$$

Our truncated steepest descent methods, shown below, arise from minimizing this truncated model either with a norm ball constraint or with squared norm regularization.

**Proposition 4.1.** [Constrained Momo] The ball constrained truncated model update is given by

$$\boldsymbol{w}_{t+1} = \underset{\|\boldsymbol{w}-\boldsymbol{w}_t\|\le\eta}{\arg\min} \left\{ \max\left(\tilde{F}_t + \langle \boldsymbol{m}_t, \boldsymbol{w}-\boldsymbol{w}_t\rangle, F_*\right) \right\} \tag{21}$$

$$= \boldsymbol{w}_t + \min\left(\eta, \frac{\tilde{F}_t - F_*}{\|\boldsymbol{m}_t\|_*}\right) \mathsf{LMO}(\boldsymbol{m}_t) \tag{22}$$

The $\arg\min$ above can have multiple solutions: we take the one that has minimal distance to $\boldsymbol{w}_t$.

**Proposition 4.2.** [Regularized Momo] The regularized truncated model update is given by

$$\boldsymbol{w}_{t+1} = \underset{\boldsymbol{w}}{\arg\min} \left\{ \max\left(\tilde{F}_t + \langle \boldsymbol{m}_t, \boldsymbol{w}-\boldsymbol{w}_t\rangle, F_*\right) + \frac{1}{2\eta}\|\boldsymbol{w}-\boldsymbol{w}_t\|^2 \right\} \tag{23}$$

$$= \boldsymbol{w}_t + \min\left(\eta, \frac{\tilde{F}_t - F_*}{\|\boldsymbol{m}_t\|_*^2}\right) \|\boldsymbol{m}_t\|_* \mathsf{LMO}(\boldsymbol{m}_t) \tag{24}$$

The term $\tilde{F}_t$ relies on the history of previous gradients and losses, but it can be computed with a single scalar running average. Pseudocode for both Momo variations is shown in Algorithm 2.

Now that we have shown how to use Momo with respect to any norm, we can immediately combine Momo with any steepest descent algorithm in our framework, including MuonAdam. For example, our proposed algorithm MuonMax-Momo (Algorithm 3 in Appendix B) can be written as Regularized Momo w.r.t. $\|\cdot\|_{\mathrm{MM}}$ (defined in Equation 17) with stale dual norm approximations.

**Proposition 4.3.** [MuonMax-Momo] Regularized Momo with respect to the norm $\|\cdot\|_{\mathrm{MM}}$ as defined in equation 17 has the following closed form:

$$d_t = \sqrt{\left(\sum_{\ell=1}^{L}\|\boldsymbol{M}_t^\ell\|_{\mathrm{nuc}}\right)^2 + \frac{\eta_b}{\eta_m}\left\|\frac{\boldsymbol{m}_t^\theta}{\sqrt{\sqrt{\boldsymbol{v}_t^\theta}+\epsilon}}\right\|_2^2}$$

$$\boldsymbol{W}_{t+1}^\ell = \boldsymbol{W}_t^\ell - \min\left\{\eta_m, \frac{\tilde{F}_t - F_*}{d_t^2}\right\}\left(\sum_{j=1}^{L}\|\boldsymbol{M}_t^j\|_{\mathrm{nuc}}\right)\mathrm{polar}(\boldsymbol{M}_t^\ell) \tag{25}$$

$$\boldsymbol{\theta}_{t+1} = \boldsymbol{\theta}_t - \min\left\{\eta_b, \frac{\eta_b}{\eta_m}\frac{\tilde{F}_t - F_*}{d_t^2}\right\}\frac{\boldsymbol{m}_t^\theta}{\sqrt{\boldsymbol{v}_t^\theta}+\epsilon}.$$

The update in Proposition 4.3 matches that of Algorithm 3 except for the use of stale dual norms.

## 5 EXPERIMENTS

Here we provide a comprehensive evaluation of optimizers arising from our steepest descent framework for training language models. We start by tuning and evaluating 36 optimizer variations arising from different choices of normalization, product norm, norm for the non-matrix parameters, and whether to use model truncation. For this initial phase of evaluating all variations, we use 1B tokens from the FineWeb dataset to train a GPT2-Small model with 124M params (Section 5.1). We take the four best performing methods (MuonAdam, Scion, MuonAdam-Momo, MuonMax-Momo) and evaluate them for a GPT2-Large model with 774M params on the SlimPajama dataset (Section 5.2), first by thoroughly tuning all four algorithms with 1B tokens, then running a final evaluation of Muon and MuonMax-Momo with 6B tokens. Finally, in Section 5.3 we perform two ablation studies: we examine the sensitivity of Momo variants to the choice of the loss lower bound $F_*$, and we evaluate the effect of stale nuclear norm approximations on final loss and wall-clock time.

### 5.1 FINEWEB DATASET

To identify the strongest methods within our framework, we thoroughly tune and evaluate 36 variations that arise from mixing and matching settings for the following design choices: constrained vs regularized steepest descent, product norm ($\|\cdot\|_\infty, \|\cdot\|_2, \|\cdot\|_{\mathrm{hyb}}$), norm for parameters besides hidden weight matrices ($\|\cdot\|_\infty, \|\cdot\|_{\mathrm{ada}\infty}, \|\cdot\|_{\mathrm{ada}2}$), and whether to apply model truncation.

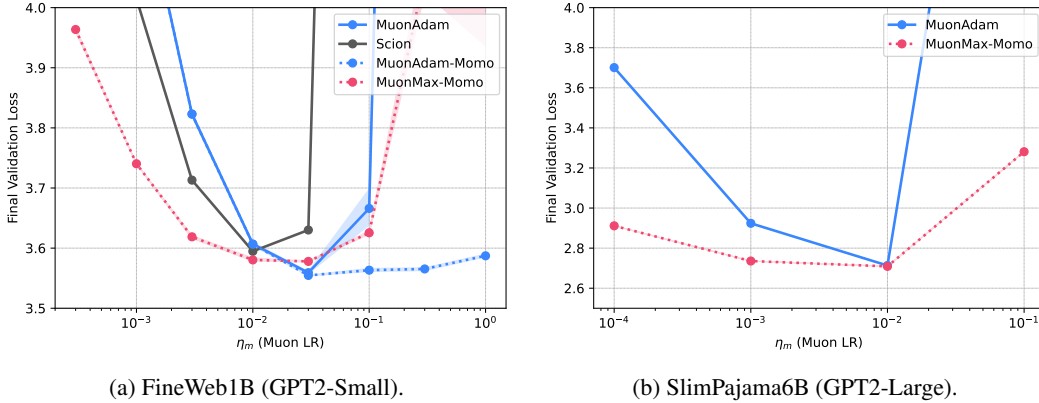

(a) FineWeb1B (GPT2-Small).  (b) SlimPajama6B (GPT2-Large).

Figure 2: Final validation loss with varying learning rates on FineWeb1B (left) and SlimPajama6B (right). Our MuonAdam-Momo and MuonMax-Momo have wider basins than MuonAdam and Scion, indicating increased robustness to learning rate tuning.

**Setup.** For all variations, we run one epoch of training with 1B tokens from the FineWeb dataset (Penedo et al., 2024), using the GPT-2 Small architecture (124M params) from modded-nanogpt (Jordan et al., 2024a). Each algorithm in our framework has two learning rates: $\eta_m$ for the hidden weight matrices (which we call the Muon learning rate) and $\eta_b$ for everything else (which we call the base learning rate). Due to the computational cost of performing a double grid search, we opt to tune with an iterated grid search; for each algorithm, we fix $\eta_b$ while tuning $\eta_m$, then fix $\eta_m$ at the tuned value while tuning $\eta_b$. See Appendix C for a complete specification of the tuning protocol and other implementation details. For all Momo variations, we set the lower bound $F_* = 3.2$, and conduct a sensitivity analysis of this hyperparameter in Section 5.3.

**Results.** The final loss for each variation is shown in Tables 2 and 3 of Appendix D. For the best performing variations (MuonAdam, Scion, MuonAdam-Momo, and MuonMax-Momo), we additionally evaluate the sensitivity to learning rate tuning by running each algorithm with LRs $(\rho\eta_m, \rho\eta_b)$, where $(\eta_m, \eta_b)$ are the previously tuned LRs and $\rho$ varies over $\{0.03, 0.1, 0.3, 1, 3, 10, 30, 100\}$, with three random seeds each (Figure 2a). Table 4 in Appendix D gives the mean and standard deviation of final validation loss for each algorithm with tuned LRs. For these runs, MuonAdam-Momo and MuonMax-Momo use stale nuclear norms.

In Figure 2a, we see that MuonAdam and MuonAdam-Momo achieve the smallest loss among all variations, though MuonAdam is much more sensitive to the learning rate. Both MuonAdam-Momo and MuonMax-Momo enjoy a much wider range of competitive learning rates compared with MuonAdam and Scion; for this search range, the proportion of LRs yielding loss less than 3.65 is 25% for MuonAdam and Scion, 50% for MuonMax-Momo, and 62.5% for MuonAdam-Momo. Also, Table 4 (Appendix D) shows that both of our Momo methods achieve a smaller variation in loss across random seeds compared with MuonAdam and Scion.

## 5.2 SlimPajama Dataset

Having identified MuonAdam, Scion, MuonAdam-Momo, and MuonMax-Momo as the strongest variations, we evaluate these methods for training the GPT2-Large architecture (774M params) using the SlimPajama dataset (Soboleva et al., 2023). We first evaluate all four algorithms for one epoch with 1B tokens, then evaluate MuonAdam and MuonMax-Momo for one epoch with 6B tokens.

**Setup.** Most aspects of training are the same as in Section 5.1, the main difference being the tuning protocol. To tune the two learning rates $\eta_m$ and $\eta_b$, we run a double grid search for each algorithm, varying $\eta_m \in \{$1e-4, 1e-3, 1e-2, 1e-1$\}$ and $\eta_b \in \{$1e-5, 1e-4, 1e-3, 1e-2$\}$ for a total of 16 settings per algorithm. For the Momo variants, we set the lower bound $F_* = 2.8$ when training with 1B tokens and $F_* = 2.0$ when training with 6B tokens. We did not tune $F_*$, and based on the sensitivity analysis in Section 5.3, we expect that this hyperparameter does not have a large effect on final performance for tuned learning rates.

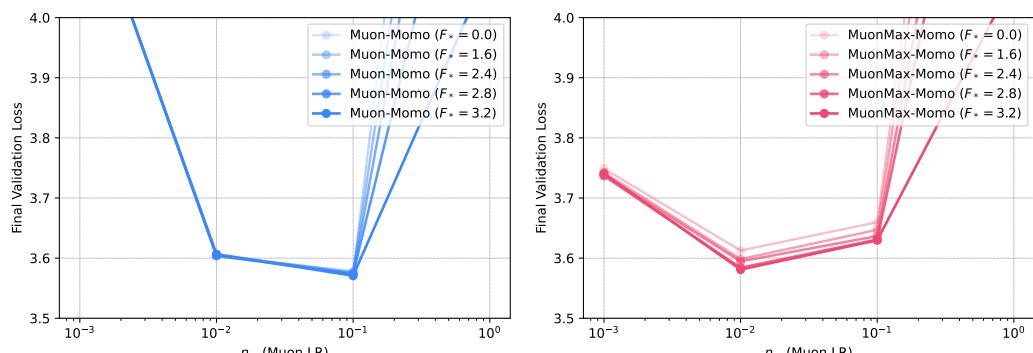

Figure 3: Sensitivity to loss lower bound $F_*$ for model truncation (Fineweb1B).

|  | MuonAdam | MuonMax | MuonAdam-Momo | Scion-Momo | MuonMax-Momo |
|---|---|---|---|---|---|
| Original | 3.604 (1×) | 3.791 (1.09×) | 3.551 (1.10×) | 3.592 (1.08×) | 3.576 (1.11×) |
| Stale | - | 3.768 (1.04×) | 3.554 (1.04×) | 3.590 (1.02×) | 3.580 (1.05×) |

Table 1: Effect of stale nuclear norm approximation on final loss and wall-clock time per-iteration compared to MuonAdam, which has no stale variant because it does not involve nuclear norms.

**Results.** Figure 1 shows the final loss of each method with LRs $(\rho\eta_m, \rho\eta_b)$, where $(\eta_m, \eta_b)$ are tuned LRs and $\rho \in \{1e\text{-}2, 1e\text{-}1, 1, 1e1, 1e2, 1e3\}$. The sensitivity of each method with respect to both learning rates is shown for the full 2D grid in Figure 9 of Appendix D. We see in Figure 1 that MuonMax-Momo achieves the lowest loss of all methods, and that both Momo variations are extremely robust to the choice of learning rates. Both MuonAdam and Scion have quite narrow sensitivity curves, that is, shifting the optimal learning rates by a factor of 10 in either direction creates a large increase in final loss. In comparison, the final loss of MuonMax-Momo remains between 3.13 and 3.24 even as $\eta_m$ varies over five orders of magnitude from 1e-3 to 10.

We see similar robustness of MuonMax-Momo when scaling up to 6B tokens. Due to the cost of re-tuning learning rates, we reuse the ratio $\eta_m/\eta_b$ of the tuned learning rates from 1B training, and vary $\eta_m \in \{1e\text{-}4, 1e\text{-}3, 1e\text{-}2, 1e\text{-}1\}$ for MuonAdam and MuonMax-Momo. Figure 2b shows that MuonMax-Momo achieves a lower loss than MuonAdam for every setting in this range, and generally exhibits less variation in the loss as the learning rates are shifted from their optimal values.

## 5.3 ABLATIONS

To probe the behavior of our proposed methods, we perform two ablation studies: (1) we evaluate how the choice of loss lower bound $F_*$ affects the final validation loss of MuonAdam-Momo and MuonMax-Momo; (2) we evaluate the effect of using stale nuclear norm approximations on the final validation loss and wall-clock time per iteration for several methods in our framework. In this section, we use the same setup as in Section 5.1 (GPT2-Small, FineWeb dataset, 1B tokens).

**Sensitivity Analysis of $F_*$.** Figure 3 shows the final loss of MuonAdam-Momo and MuonMax-Momo with various $\eta_m$, as the loss lower bound $F_*$ varies over $\{0, 1.6, 2.4, 2.8, 3.2\}$. We see that the choice of $F_*$ makes the biggest difference when $\eta_m$ is larger than the optimal value. For MuonAdam-Momo, the final loss is nearly identical for all values of $F_*$ when $\eta_m \leq 0.1$. MuonMax-Momo is somewhat more sensitive to the choice of $F_*$, but even the aggressive lower bound of $F_* = 0.0$ achieves 3.61 loss compared to the 3.58 optimum achieved with $F_* = 3.2$.

**Effect of Stale Approximation.** Table 1 shows the final validation loss and per-step wall-clock times of four methods (with tuned LRs) with and without stale nuclear norm approximations. We see that in all cases, the stale approximation increases the loss by at most 0.004, while sometimes even decreasing it. We therefore conclude that this approximation does not noticeably affect the final loss for these tuned algorithms, although it does afford a speedup; for MuonMax-Momo, the additional wall-clock time compared to MuonAdam is reduced from 11% to 5%.

## REPRODUCIBILITY STATEMENT

All of the code we used for our experiments is included in the supplementary material. This includes code to download and process data, run training, and make plots. See README.md in the supplementary material for instructions on running our code. On a conceptual level, all of our proposed methods are derived in full detail, and can in principle be implemented from scratch in PyTorch without any additional knowledge outside of the paper. Full pseudocode for our proposed method is given in Algorithm 3, and any of the other methods we discuss in this paper can be derived from Propositions 3.1, 3.2, 4.1, 4.2, Lemma 3.3, and the specifications in Table 2.

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

CONTENTS

## A  PROOFS FROM SECTION 3

In what follows, for a norm denoted by a subscript such as $\|\cdot\|_\infty$, we will sometimes replace $\text{LMO}_{\|\cdot\|_\infty}$ with $\text{LMO}_\infty$.

**Proposition 3.1.** [Constrained Steepest Descent] The CSD update is given by

$$\boldsymbol{w}_{t+1} = \underset{\|\boldsymbol{w}-\boldsymbol{w}_t\|\leq\eta}{\arg\min} \ \{F(\boldsymbol{w}_t) + \langle \boldsymbol{m}_t, \boldsymbol{w} - \boldsymbol{w}_t\rangle\} \ = \ \boldsymbol{w}_t + \eta\,\text{LMO}(\boldsymbol{m}_t). \tag{2}$$

*Proof of Proposition 3.1.* Denoting $r = \|\boldsymbol{w}-\boldsymbol{w}_t\|$ and $\boldsymbol{\Delta} = (\boldsymbol{w}-\boldsymbol{w}_t)/\|\boldsymbol{w}-\boldsymbol{w}_t\|$, we can change variables in the optimization problem from Equation 2, yielding $\boldsymbol{w}_{t+1} = \boldsymbol{w}_t + r_t\boldsymbol{\Delta}_t$, where

$$(r_t, \boldsymbol{\Delta}_t) = \underset{r\in[0,\eta],\|\boldsymbol{\Delta}\|=1}{\arg\min} \ \{r\langle \boldsymbol{m}_t, \boldsymbol{\Delta}\rangle\}, \tag{26}$$

which can be separated into

$$\boldsymbol{\Delta}_t = \underset{\|\boldsymbol{\Delta}\|=1}{\arg\min}\langle \boldsymbol{m}_t, \boldsymbol{\Delta}\rangle = \text{LMO}(\boldsymbol{m}_t), \tag{27}$$

and

$$r_t = \underset{r\in[0,\eta]}{\arg\min}\ \{r\langle \boldsymbol{m}_t, \boldsymbol{\Delta}_t\rangle\} = \underset{r\in[0,\eta]}{\arg\min}\ \{r\langle \boldsymbol{m}_t, \text{LMO}(\boldsymbol{m}_t)\rangle\} = \underset{r\in[0,\eta]}{\arg\min}\ \{-r\|\boldsymbol{m}_t\|_*\} = \eta, \tag{28}$$

so $\boldsymbol{w}_{t+1} = \boldsymbol{w}_t + \eta\,\text{LMO}(\boldsymbol{m}_t)$. □

**Proposition 3.2.** [Regularized Steepest Descent] The RSD update is given by

$$\boldsymbol{w}_{t+1} = \underset{\boldsymbol{w}}{\arg\min}\ \Big\{F(\boldsymbol{w}_t) + \langle \boldsymbol{m}_t, \boldsymbol{w} - \boldsymbol{w}_t\rangle + \tfrac{1}{2\eta}\|\boldsymbol{w}-\boldsymbol{w}_t\|^2\Big\} \ = \ \boldsymbol{w}_t + \eta\|\boldsymbol{m}_t\|_*\text{LMO}(\boldsymbol{m}_t) \tag{3}$$

*Proof of Proposition 3.2.* For the optimization problem from Equation 3, we use the same change of variables as in the proof of Proposition 3.1: $r = \|\boldsymbol{w}-\boldsymbol{w}_t\|$ and $\boldsymbol{\Delta} = (\boldsymbol{w}-\boldsymbol{w}_t)/\|\boldsymbol{w}-\boldsymbol{w}_t\|$. Therefore $\boldsymbol{w}_{t+1} = \boldsymbol{w}_t + r_t\boldsymbol{\Delta}_t$, where

$$(r_t, \boldsymbol{\Delta}_t) = \underset{r\geq 0,\|\boldsymbol{\Delta}\|=1}{\arg\min}\ \Big\{r\langle \boldsymbol{m}_t, \boldsymbol{\Delta}\rangle + \tfrac{r^2}{2\eta}\Big\}, \tag{29}$$

which can be separated into

$$\boldsymbol{\Delta}_t = \underset{\|\boldsymbol{\Delta}\|=1}{\arg\min}\langle \boldsymbol{m}_t, \boldsymbol{\Delta}\rangle = \text{LMO}(\boldsymbol{m}_t), \tag{30}$$

and

$$r_t = \underset{r\geq 0}{\arg\min}\ \Big\{r\langle \boldsymbol{m}_t, \boldsymbol{\Delta_t}\rangle + \tfrac{r^2}{2\eta}\Big\} \tag{31}$$

$$= \underset{r\geq 0}{\arg\min}\ \Big\{r\langle \boldsymbol{m}_t, \text{LMO}(\boldsymbol{m}_t)\rangle + \tfrac{r^2}{2\eta}\Big\} \tag{32}$$

$$= \underset{r\geq 0}{\arg\min}\ \Big\{-r\|\boldsymbol{m}_t\|_* + \tfrac{r^2}{2\eta}\Big\} \tag{33}$$

$$= \eta\|\boldsymbol{m}_t\|_*, \tag{34}$$

so that $\boldsymbol{w}_{t+1} = \boldsymbol{w}_t + \eta\|\boldsymbol{m}_t\|_*\text{LMO}(\boldsymbol{m}_t)$. □

**Lemma 3.3.** [LMO and Dual of Product Norms] For each $i \in [n]$, let $g_i$ be a norm on $\mathbb{R}^{d_i}$, and let $f$ be a norm on $\mathbb{R}^n$, and denote their dual norms as $g_{i,*}$ and $f_*$, respectively. Then the product norm $h : \mathbb{R}^{d_1} \times \ldots \times \mathbb{R}^{d_n} \to \mathbb{R}$ defined by

$$h(\boldsymbol{w}^1, \ldots, \boldsymbol{w}^n) = f\big(g_1(\boldsymbol{w}^1), \ldots, g_n(\boldsymbol{w}^n)\big) \tag{5}$$

is indeed a norm, and its LMO and dual norm are given by

$$\mathsf{LMO}_h(\boldsymbol{w}^1, \ldots, \boldsymbol{w}^n) = (\phi_1 \mathsf{LMO}_{g_1}(\boldsymbol{w}^1), \ldots, \phi_n \mathsf{LMO}_{g_n}(\boldsymbol{w}^n)) \tag{6}$$

$$h_*(\boldsymbol{w}^1, \ldots, \boldsymbol{w}^n) = f_*(g_{1,*}(\boldsymbol{w}^1), \ldots, g_{n,*}(\boldsymbol{w}^n)), \tag{7}$$

where $(\phi_1, \ldots, \phi_n) := -\mathsf{LMO}_f(g_{1,*}(\boldsymbol{w}^1), \ldots, g_{n,*}(\boldsymbol{w}^n))$.

*Proof of Lemma 3.3.* To show that $h$ is a norm, we only need to show that

1. $h(\boldsymbol{w}_1, \ldots, \boldsymbol{w}_n) \geq 0$ for all $\boldsymbol{w}_1, \ldots, \boldsymbol{w}_n$,

2. $h(\boldsymbol{w}_1, \ldots, \boldsymbol{w}_n) = 0$ if and only if $(\boldsymbol{w}_1 \ldots, \boldsymbol{w}_n) = \boldsymbol{0}$,

3. $h(\lambda \boldsymbol{w}_1, \ldots, \lambda \boldsymbol{w}_n) = |\lambda| h(\boldsymbol{w}_1, \ldots, \boldsymbol{w}_n)$ for all $\lambda \in \mathbb{R}$, $\boldsymbol{w}_1, \ldots, \boldsymbol{w}_n$,

4. $h(\boldsymbol{w}_1 + \boldsymbol{v}_1, \ldots, \boldsymbol{w}_n + \boldsymbol{v}_n) \leq h(\boldsymbol{w}_1, \ldots, \boldsymbol{w}_n) + h(\boldsymbol{v}_1, \ldots, \boldsymbol{v}_n)$ for all $\boldsymbol{w}_1, \boldsymbol{v}_1, \ldots, \boldsymbol{w}_n, \boldsymbol{v}_n$.

All of these properties hold immediately from the definition of $h = f \circ (g_1, \ldots, g_n)$ together with repeated applications of the norm properties of $f$ and $g_1, \ldots, g_n$.

From the definition of the dual norm,

$$h_*(\boldsymbol{w}_1, \ldots, \boldsymbol{w}_n) = \max \left\{ \sum_{i=1}^n \langle \boldsymbol{w}_i, \boldsymbol{v}_i \rangle \,\middle|\, h(\boldsymbol{v}_1, \ldots, \boldsymbol{v}_n) = 1 \right\} \tag{35}$$

$$= \max \left\{ \sum_{i=1}^n \langle \boldsymbol{w}_i, \boldsymbol{v}_i \rangle \,\middle|\, f(g_1(\boldsymbol{v}_1), \ldots, g_n(\boldsymbol{v}_n)) = 1 \right\}. \tag{36}$$

We use a change of variables $\boldsymbol{u}_i = \boldsymbol{v}_i / g_i(\boldsymbol{v}_i)$ and $r_i = g_i(\boldsymbol{v}_i)$, which separates the update direction $\boldsymbol{u}_i$ (with unit norm) from the update norm $r_i$. So Equation 36 is equivalent to

$$h_*(\boldsymbol{w}_1, \ldots, \boldsymbol{w}_n) = \max \left\{ \sum_{i=1}^n r_i \langle \boldsymbol{w}_i, \boldsymbol{u}_i \rangle \,\middle|\, f(r_1, \ldots, r_n) = 1 \right\} \tag{37}$$

Note that the condition $f(r_1, \ldots, r_n) = 1$ does not involve $\boldsymbol{u}_i$, so each term $r_i \langle \boldsymbol{w}_i, \boldsymbol{u}_i \rangle$ is maximized when

$$\boldsymbol{u}_i = \arg\max_{g_i(\boldsymbol{z}_i)=1} \langle \boldsymbol{w}_i, \boldsymbol{z}_i \rangle = -\mathsf{LMO}_{g_i}(\boldsymbol{w}_i), \tag{38}$$

with maximum value $\langle \boldsymbol{w}_i, \boldsymbol{u}_i \rangle = g_{i,*}(\boldsymbol{w}_i)$. Using this in Equation 37 gives

$$h_*(\boldsymbol{w}_1, \ldots, \boldsymbol{w}_n) = \max \left\{ \sum_{i=1}^n r_i g_{i,*}(\boldsymbol{w}_i) \,\middle|\, f(r_1, \ldots, r_n) = 1 \right\}. \tag{39}$$

Denoting $\boldsymbol{r} = (r_1, \ldots, r_n)$ and $\boldsymbol{s} = (g_{1,*}(\boldsymbol{w}_1), \ldots, g_{n,*}(\boldsymbol{w}_n))$, this is equivalent to

$$h_*(\boldsymbol{w}_1, \ldots, \boldsymbol{w}_n) = \max \left\{ \langle \boldsymbol{r}, \boldsymbol{s} \rangle \mid f(\boldsymbol{r}) = 1 \right\} \tag{40}$$

$$= f_*(\boldsymbol{s}), \tag{41}$$

which gives us the dual norm $h_*$.

To obtain $\mathsf{LMO}_h$, we only need to look at the value of the variables which achieved the maximum in the above derivation:

$$\boldsymbol{u}_i = -\mathsf{LMO}_{g_i}(\boldsymbol{w}_i), \quad \text{and} \quad \boldsymbol{r} = \mathsf{LMO}_f(g_{1,*}(\boldsymbol{w}_1), \ldots, g_{n,*}(\boldsymbol{w}_n)) \tag{42}$$

so that

$$\boldsymbol{v}_i = r_i \mathsf{LMO}_f(g_{1,*}(\boldsymbol{w}_1), \ldots, g_{n,*}(\boldsymbol{w}_n)) \tag{43}$$

maximizes $\sum_{i=1}^n \langle \boldsymbol{w}_i, \boldsymbol{v}_i \rangle$ subject to $h(\boldsymbol{v}_1, \ldots, \boldsymbol{v}_n) = 1$. Note that $\mathsf{LMO}_h(\boldsymbol{w}_1, \ldots, \boldsymbol{w}_n)$ is exactly the minimizer of $\sum_{i=1}^n \langle \boldsymbol{w}_i, \boldsymbol{v}_i \rangle$ subject to the same norm constraint; since $\sum_{i=1}^n \langle \boldsymbol{w}_i, \boldsymbol{v}_i \rangle$ is linear in $\boldsymbol{v}_i$, the minimizer is the negative of the maximizer. Therefore

$$\mathsf{LMO}_h(\boldsymbol{w}_1, \ldots, \boldsymbol{w}_n) = -(r_1 \mathsf{LMO}_{g_1}(\boldsymbol{w}_1), \ldots, r_n \mathsf{LMO}_{g_n}(\boldsymbol{w}_n)). \tag{44}$$

□

The following lemma will be useful later for quickly computing duals and LMOs of various norms.

**Lemma A.1.** For any norm $\|\cdot\|$ on $\mathbb{R}^d$ full rank matrix $\boldsymbol{D} \in \mathbb{R}^{d \times d}$, the norm defined by $\|\boldsymbol{v}\|_{\boldsymbol{D}} = \|\boldsymbol{D}\boldsymbol{v}\|$ has

$$\mathsf{LMO}_{\|\cdot\|_{\boldsymbol{D}}}(\boldsymbol{v}) = \boldsymbol{D}^{-1}\mathsf{LMO}_{\|\cdot\|}(\boldsymbol{D}^{-T}\boldsymbol{v}), \tag{45}$$

$$\|\boldsymbol{v}\|_{\boldsymbol{D},*} = \|\boldsymbol{D}^{-T}\boldsymbol{v}\|_*. \tag{46}$$

*Proof.* The fact that $\|\cdot\|_{\boldsymbol{D}}$ is a norm follows immediately from the norm properties of $\|\cdot\|$ together with the fact that $\boldsymbol{D}$ is full rank. For the dual norm,

$$\|\boldsymbol{v}\|_{\boldsymbol{D},*} = \max_{\|\boldsymbol{u}\|_{\boldsymbol{D}}=1} \langle \boldsymbol{v}, \boldsymbol{u} \rangle = \max_{\|\boldsymbol{D}\boldsymbol{u}\|=1} \langle \boldsymbol{v}, \boldsymbol{u} \rangle \tag{47}$$

and a change of variables $\boldsymbol{z} = \boldsymbol{D}\boldsymbol{u}$ yields

$$\|\boldsymbol{v}\|_{\boldsymbol{D},*} = \max_{\|\boldsymbol{z}\|=1} \langle \boldsymbol{v}, \boldsymbol{D}^{-1}\boldsymbol{z} \rangle = \max_{\|\boldsymbol{z}\|=1} \langle \boldsymbol{D}^{-T}\boldsymbol{v}, \boldsymbol{z} \rangle = \|\boldsymbol{D}^{-T}\boldsymbol{v}\|_*. \tag{48}$$

For the LMO, we can look at the value of the variables that maximize the inner product in the above:

$$\boldsymbol{z} = \arg\max_{\|\boldsymbol{z}\|=1} \langle \boldsymbol{D}^{-T}\boldsymbol{v}, \boldsymbol{z} \rangle = -\mathsf{LMO}_{\|\cdot\|}(\boldsymbol{D}^{-T}\boldsymbol{v}). \tag{49}$$

Returning to the $\boldsymbol{u}$ variable then gives

$$\boldsymbol{u} = \boldsymbol{D}^{-1}\boldsymbol{z} = -\boldsymbol{D}^{-1}\mathsf{LMO}_{\|\cdot\|}(\boldsymbol{D}^{-T}\boldsymbol{v})$$

which maximizes $\langle \boldsymbol{v}, \boldsymbol{u} \rangle$ subject to $\|\boldsymbol{u}\|_{\boldsymbol{D}} = 1$. Since $\langle \boldsymbol{v}, \boldsymbol{u} \rangle$ is linear in $\boldsymbol{u}$, the minimizer of $\langle \boldsymbol{v}, \boldsymbol{u} \rangle$ under the norm constraint $\|\boldsymbol{u}\|_{\boldsymbol{D}} = 1$ is exactly the negative of the maximizer under the same constraint. So

$$\mathsf{LMO}_{\|\cdot\|_{\boldsymbol{D}}}(\boldsymbol{v}) = \arg\min_{\|\boldsymbol{u}\|_{\boldsymbol{D}}=1} \langle \boldsymbol{v}, \boldsymbol{u} \rangle = \boldsymbol{D}^{-1}\mathsf{LMO}_{\|\cdot\|}(\boldsymbol{D}^{-T}\boldsymbol{v}) \tag{50}$$

□

**Proposition 3.4.** The $t$-th update of Adam is the CSD with step size $\eta$ with respect to the norm:

$$\|\boldsymbol{\theta}\|_{\text{ada}\infty} := \left\|\text{Diag}\left(\frac{\sqrt{\boldsymbol{v}_t}+\epsilon}{|\boldsymbol{m}_t|}\right)\boldsymbol{\theta}\right\|_\infty \tag{13}$$

*Proof of Proposition 3.4.* Let $\boldsymbol{D} = \text{Diag}\left(\frac{\sqrt{\boldsymbol{v}_t}+\epsilon}{|\boldsymbol{m}_t|}\right)$, so that $\|\boldsymbol{\theta}\|_{\text{ada}\infty} = \|\boldsymbol{D}\boldsymbol{\theta}\|_\infty$. Then by Proposition 3.1, one step of CSD w.r.t. $\|\cdot\|_{\text{ada}\infty}$ is given by

$$\boldsymbol{\theta}_{t+1} = \boldsymbol{\theta}_t + \eta\mathsf{LMO}_{\text{ada}\infty}(\boldsymbol{m}_t) \tag{51}$$

$$\overset{(i)}{=} \boldsymbol{\theta}_t + \eta\boldsymbol{D}^{-1}\mathsf{LMO}_\infty(\boldsymbol{D}^{-T}\boldsymbol{m}_t) \tag{52}$$

$$\overset{(ii)}{=} \boldsymbol{\theta}_t - \eta\boldsymbol{D}^{-1}\text{sign}(\boldsymbol{D}^{-T}\boldsymbol{m}_t) \tag{53}$$

$$= \boldsymbol{\theta}_t - \eta\text{Diag}\left(\frac{|\boldsymbol{m}_t|}{\sqrt{\boldsymbol{v}_t}+\epsilon}\right)\text{sign}\left(\text{Diag}\left(\frac{|\boldsymbol{m}_t|}{\sqrt{\boldsymbol{v}_t}+\epsilon}\right)\boldsymbol{m}_t\right) \tag{54}$$

$$= \boldsymbol{\theta}_t - \eta\frac{|\boldsymbol{m}_t|}{\sqrt{\boldsymbol{v}_t}+\epsilon} \odot \text{sign}(\boldsymbol{m}_t) \tag{55}$$

$$= \boldsymbol{\theta}_t - \eta\frac{\boldsymbol{m}_t}{\sqrt{\boldsymbol{v}_t}+\epsilon}, \tag{56}$$

where $(i)$ uses Lemma A.1 and $(ii)$ uses $\mathsf{LMO}_\infty(\boldsymbol{v}) = -\text{sign}(\boldsymbol{v})$. □

**Proposition 3.5.** The $t$-th update of Adam is the RSD with step size $\eta$ with respect to the norm:

$$\|\boldsymbol{\theta}\|_{\text{ada}2} := \sqrt{\langle\text{Diag}(\sqrt{\boldsymbol{v}_t}+\epsilon)\boldsymbol{\theta}, \boldsymbol{\theta}\rangle} = \left\|\text{Diag}\left(\sqrt{\sqrt{\boldsymbol{v}_t}+\epsilon}\right)\boldsymbol{\theta}\right\|_2 \tag{14}$$

---

**Algorithm 1** MuonAdam: where $\boldsymbol{W}^1, \ldots, \boldsymbol{W}^L$ are the weight matrices, and $\boldsymbol{\theta}$ are all other parameters flattened into a vector.

---

**Inputs:** $\boldsymbol{W}_0^1, \ldots, \boldsymbol{W}_0^L, \boldsymbol{\theta}_0$, learning rates $\eta_b, \eta_m$, EMA parameters $\beta, \beta_1, \beta_2$

1: **for** $t = 0, 1, \ldots, T-1$ **do**
2:    $(\boldsymbol{G}_t^1, \ldots, \boldsymbol{G}_t^L, \boldsymbol{g}_t^\theta) \leftarrow \text{backward}(\boldsymbol{W}_t^1, \ldots \boldsymbol{W}_t^L, \boldsymbol{\theta}_t)$
3:    **for** $\ell = 1, \ldots, L$ **do**
4:       $\boldsymbol{M}_t^\ell = \beta \boldsymbol{M}_{t-1}^\ell + (1-\beta)\boldsymbol{G}_t^\ell$
5:       $\boldsymbol{W}_{t+1}^\ell \leftarrow \boldsymbol{W}_t^\ell - \eta_m \text{polar}(\boldsymbol{M}_t^\ell)$
6:    **end for**
7:    $\boldsymbol{m}_t^\theta = \beta_1 \boldsymbol{m}_{t-1}^\theta + (1-\beta_1)\boldsymbol{g}_t^\theta$
8:    $\boldsymbol{v}_t^\theta = \beta_2 \boldsymbol{v}_{t-1}^\theta + (1-\beta_2)\boldsymbol{g}_t^\theta \odot \boldsymbol{g}_t^\theta$
9:    $\boldsymbol{\theta}_{t+1} = \boldsymbol{\theta}_t - \eta_b \dfrac{\boldsymbol{m}_t^\theta}{\sqrt{\boldsymbol{v}_t^\theta} + \epsilon}$
10: **end for**

---

*Proof of Proposition 3.5.* Let $\boldsymbol{D} = \text{Diag}\left(\sqrt{\sqrt{\boldsymbol{v}_t} + \epsilon}\right)$, so that $\|\boldsymbol{\theta}\|_{\text{ada2}} = \|\boldsymbol{D}\boldsymbol{\theta}\|_2$. Then by Proposition 3.2, one step of RSD w.r.t. $\|\cdot\|_{\text{ada2}}$ is given by

$$\boldsymbol{\theta}_{t+1} = \boldsymbol{\theta}_t + \eta\|\boldsymbol{m}_t\|_{\text{ada2},*}\text{LMO}_{\text{ada2}}(\boldsymbol{m}_t) \tag{57}$$

$$\overset{(i)}{=} \boldsymbol{\theta}_t + \eta\|\boldsymbol{D}^{-T}\boldsymbol{m}_t\|_{2,*}\boldsymbol{D}^{-1}\text{LMO}_2(\boldsymbol{D}^{-T}\boldsymbol{m}_t) \tag{58}$$

$$\overset{(ii)}{=} \boldsymbol{\theta}_t - \eta\|\boldsymbol{D}^{-T}\boldsymbol{m}_t\|_2\boldsymbol{D}^{-1}\frac{\boldsymbol{D}^{-T}\boldsymbol{m}_t}{\|\boldsymbol{D}^{-T}\boldsymbol{m}_t\|_2} \tag{59}$$

$$= \boldsymbol{\theta}_t - \eta\boldsymbol{D}^{-1}\boldsymbol{D}^{-T}\boldsymbol{m}_t \tag{60}$$

$$= \boldsymbol{\theta}_t - \eta\text{Diag}\left(\frac{1}{\sqrt{\boldsymbol{v}_t}+\epsilon}\right)\boldsymbol{m}_t \tag{61}$$

$$= \boldsymbol{\theta}_t - \eta\frac{\boldsymbol{m}_t}{\sqrt{\boldsymbol{v}_t}+\epsilon}, \tag{62}$$

where $(i)$ uses Lemma A.1 and $(ii)$ uses $\text{LMO}_2(\boldsymbol{v}) = -\boldsymbol{v}/\|\boldsymbol{v}\|_2$. $\qquad\square$

For reference, we include the pseudocode for MuonAdam (Muon side-by-side with Adam) in Algorithm 1.

**Proposition 3.6.** MuonAdam (Algorithm 1) is exactly CSD with step size $\eta_m$ with respect to

$$\|\boldsymbol{W}\|_{\text{muon}} = \max\left(\max_{\ell \in [L]}\|\boldsymbol{W}^\ell\|_{2\to2}, \frac{\eta_m}{\eta_b}\|\boldsymbol{\theta}\|_{\text{ada}\infty}\right). \tag{15}$$

*Proof of Proposition 3.6.* By Proposition 3.1, one step of CSD w.r.t. $\|\cdot\|_{\text{muon}}$ can be written as

$$\boldsymbol{W}_{t+1} = \boldsymbol{W}_t + \eta_m\text{LMO}_{\text{muon}}(\boldsymbol{M}_t), \tag{63}$$

where $\boldsymbol{M}_t$ is the momentum buffer for all network parameters, i.e. it is the concatenation of the momentum buffers of each parameter:

$$\boldsymbol{M}_t = (\boldsymbol{M}_t^1, \ldots, \boldsymbol{M}_t^L, \boldsymbol{m}_t^\theta). \tag{64}$$

Denote $\lambda = \eta_b/\eta_m$. To compute the LMO term, we can rewrite $\|\boldsymbol{W}\|_{\text{muon}}$ as

$$\|\boldsymbol{W}\|_{\text{muon}} = \max\left(\|\boldsymbol{W}^1\|_{2\to2}, \ldots \|\boldsymbol{W}^L\|_{2\to2}, \frac{1}{\lambda}\|\boldsymbol{\theta}\|_{\text{ada}\infty}\right), \tag{65}$$

so that $\|\cdot\|_{\text{muon}}$ can be written as the composition (as in Lemma 3.3)

$$\|\boldsymbol{W}\|_{\text{muon}} = f(g_1(\boldsymbol{W}^1), \ldots g_L(\boldsymbol{W}^L), g_{L+1}(\boldsymbol{\theta})), \tag{66}$$

with $g_i(\boldsymbol{M}) = \|\boldsymbol{M}\|_{2\to2}$ for $i \in [L]$, $g_{L+1}(\boldsymbol{\theta}) = \|\boldsymbol{\theta}\|_{\text{ada}\infty}$, and $f(\boldsymbol{v}) = \|\boldsymbol{D}\boldsymbol{v}\|_\infty$, where $\boldsymbol{D} = \text{Diag}(1, \ldots, 1, 1/\lambda) \in \mathbb{R}^{(L+1)\times(L+1)}$. Therefore, by Lemma 3.3, the update in Equation 63 is equivalent to

$$\boldsymbol{W}_{t+1}^\ell = \boldsymbol{W}_t^\ell + \eta_m\phi_\ell\text{LMO}_{2\to2}(\boldsymbol{M}_t^\ell)$$
$$\boldsymbol{\theta}_{t+1} = \boldsymbol{\theta}_t + \eta_m\phi_{L+1}\text{LMO}_{\text{ada}\infty}(\boldsymbol{m}_t^\theta), \tag{67}$$

where $\phi = -\mathsf{LMO}_f(\|\boldsymbol{M}_t^1\|_{\mathrm{nuc}}, \ldots, \|\boldsymbol{M}_t^L\|_{\mathrm{nuc}}, \|\boldsymbol{m}_t^\theta\|_{\mathrm{ada}\infty,*})$. We know $\mathsf{LMO}_{2\to2}(\boldsymbol{M}) = -\mathrm{polar}(\boldsymbol{M})$, and we proved in Proposition 3.4 that

$$\mathsf{LMO}_{\mathrm{ada}\infty}(\boldsymbol{v}) = -\frac{|\boldsymbol{m}_t^\theta|}{\sqrt{\boldsymbol{v}_t^\theta}+\epsilon}\,\mathrm{sign}(\boldsymbol{v}), \tag{68}$$

so the LMO terms in Equation 67 can be simplified as

$$\begin{aligned}
\boldsymbol{W}_{t+1}^\ell &= \boldsymbol{W}_t^\ell - \eta_m \phi_\ell \mathrm{polar}(\boldsymbol{M}_t^\ell) \\
\boldsymbol{\theta}_{t+1} &= \boldsymbol{\theta}_t - \eta_m \phi_{L+1} \frac{\boldsymbol{m}_t^\theta}{\sqrt{\boldsymbol{v}_t^\theta}+\epsilon}.
\end{aligned} \tag{69}$$

To simplify $\phi$, we use Lemma A.1. Denoting $\boldsymbol{u} = (\|\boldsymbol{M}_t^1\|_{\mathrm{nuc}}, \ldots, \|\boldsymbol{M}_t^L\|_{\mathrm{nuc}}, \|\boldsymbol{m}_t^\theta\|_{\mathrm{ada}\infty,*})$, we have

$$\phi = -\mathsf{LMO}_f(\boldsymbol{u}) = -\boldsymbol{D}^{-1}\mathsf{LMO}_\infty(\boldsymbol{D}^{-T}\boldsymbol{u}) = \boldsymbol{D}^{-1}\mathrm{sign}(\boldsymbol{D}^{-T}\boldsymbol{u}) = \boldsymbol{D}^{-1}\mathbf{1}, \tag{70}$$

so that $\phi_\ell = 1$ for $\ell \in [L]$ and $\phi_{L+1} = \lambda = \eta_b/\eta_m$. Plugging back to Equation 69 gives

$$\begin{aligned}
\boldsymbol{W}_{t+1}^\ell &= \boldsymbol{W}_t^\ell - \eta_m \mathrm{polar}(\boldsymbol{M}_t^\ell) \\
\boldsymbol{\theta}_{t+1} &= \boldsymbol{\theta}_t - \eta_a \frac{\boldsymbol{m}_t^\theta}{\sqrt{\boldsymbol{v}_t^\theta}+\epsilon},
\end{aligned} \tag{71}$$

which is exactly the update from Algorithm 1. $\qquad\square$

## A.1 RECOVERING EXISTING ALGORITHMS

Propositions A.2 and A.3 below show how Scion (Pethick et al., 2025) and PolarGrad (Lau et al., 2025) are both instances of our steepest descent framework. All notation in this section follows that of Section 3.

Throughout our paper, Scion refers to the following algorithm:

$$\begin{aligned}
\boldsymbol{W}_{t+1}^\ell &= \boldsymbol{W}_t^\ell - \eta_m \mathrm{polar}(\boldsymbol{M}_t^\ell) \\
\boldsymbol{\theta}_{t+1} &= \boldsymbol{\theta}_t - \eta_b \mathrm{sign}(\boldsymbol{m}_t^\theta).
\end{aligned} \tag{72}$$

This update differs slightly from the algorithm proposed by Pethick et al. (2025) in that for each parameter matrix $\boldsymbol{W}$ of shape $d_{\mathrm{out}} \times d_{\mathrm{in}}$, we omit a coefficient of $\sqrt{d_{\mathrm{out}}/d_{\mathrm{in}}}$ from the update. This corresponds to assigning to each weight matrix the spectral norm $\|\cdot\|_{2\to2}$ rather than the RMS to RMS operator norm used by Pethick et al. (2025). Indeed, the motivation of the RMS to RMS norm is to allow for hyperparameter transfer across architecture sizes, but in our work we focus on LR sensitivity for a fixed architecture, so for simplicity we did not employ this RMS scaling. However, we could easily recover the RMS variant by replacing the spectral norm $\|\cdot\|_{2\to2}$ with the RMS to RMS operator norm.

**Proposition A.2.** Scion is exactly CSD with step size $\eta_m$ with respect to

$$\|\boldsymbol{W}\|_{\mathrm{scion}} = \max\left(\max_{1\le\ell\le L}\|\boldsymbol{W}^\ell\|_{2\to2}, \frac{\eta_m}{\eta_b}\|\boldsymbol{\theta}\|_\infty\right). \tag{73}$$

Note that the same conclusion was already reached by Pethick et al. (2025), that is, they already described Scion in terms of a norm on the space of all parameters (see their Equation (6)). We include Proposition A.2 to specify how Scion is a special case of our framework.

*Proof.* The proof is very similar to that of Proposition 3.6, since Muon-Adam differs from Scion only in that Adam is used for non-matrix parameters instead of sign SGD with momentum.

By Proposition 3.1, one step of CSD w.r.t. $\|\cdot\|_{\mathrm{scion}}$ can be written as

$$\boldsymbol{W}_{t+1} = \boldsymbol{W}_t + \eta_m \mathsf{LMO}_{\mathrm{scion}}(\boldsymbol{M}_t), \tag{74}$$

where $\boldsymbol{M}_t$ is the momentum buffer for all network parameters, i.e. it is the concatenation of momentum buffers of each parameter:

$$\boldsymbol{M}_t = (\boldsymbol{M}_t^1, \ldots, \boldsymbol{M}_t^L, \boldsymbol{m}_t^\theta). \tag{75}$$

Denote $\lambda = \eta_b/\eta_m$. To compute the LMO term, we can rewrite $\|\boldsymbol{W}\|_{\text{scion}}$ as

$$\|\boldsymbol{W}\|_{\text{scion}} = \max\left(\|\boldsymbol{W}^1\|_{2\to2}, \dots \|\boldsymbol{W}^L\|_{2\to2}, \tfrac{1}{\lambda}\|\boldsymbol{\theta}\|_\infty\right), \tag{76}$$

so that $\|\cdot\|_{\text{scion}}$ can be written as the composition (as in Lemma 3.3)

$$\|\boldsymbol{W}\|_{\text{scion}} = f(g_1(\boldsymbol{W}^1), \dots g_L(\boldsymbol{W}^L), g_{L+1}(\boldsymbol{\theta})), \tag{77}$$

with $g_i(\boldsymbol{M}) = \|\boldsymbol{M}\|_{2\to2}$ for $i \in [L]$, $g_{L+1}(\boldsymbol{\theta}) = \|\boldsymbol{\theta}\|_\infty$, and $f(\boldsymbol{v}) = \|\boldsymbol{Dv}\|_\infty$, where $\boldsymbol{D} = \text{Diag}(1, \dots, 1, 1/\lambda) \in \mathbb{R}^{(L+1)\times(L+1)}$. Therefore, by Lemma 3.3, the update in Equation 74 is equivalent to

$$\boldsymbol{W}_{t+1}^\ell = \boldsymbol{W}_t^\ell + \eta_m\phi_\ell\text{LMO}_{2\to2}(\boldsymbol{M}_t^\ell)$$
$$\boldsymbol{\theta}_{t+1} = \boldsymbol{\theta}_t + \eta_m\phi_{L+1}\text{LMO}_\infty(\boldsymbol{m}_t^\theta), \tag{78}$$

where $\boldsymbol{\phi} = -\text{LMO}_f(\|\boldsymbol{M}_t^1\|_{\text{nuc}}, \dots, \|\boldsymbol{M}_t^L\|_{\text{nuc}}, \|\boldsymbol{m}_t^\theta\|_{\text{ada}\infty,*})$. We know $\text{LMO}_{2\to2}(\boldsymbol{M}) = -\text{polar}(\boldsymbol{M})$ and $\text{LMO}_\infty(\boldsymbol{v}) = -\text{sign}(\boldsymbol{v})$, so the LMO terms in Equation 78 can be simplified as

$$\boldsymbol{W}_{t+1}^\ell = \boldsymbol{W}_t^\ell - \eta_m\phi_\ell\text{polar}(\boldsymbol{M}_t^\ell)$$
$$\boldsymbol{\theta}_{t+1} = \boldsymbol{\theta}_t - \eta_m\phi_{L+1}\text{sign}(\boldsymbol{m}_t^\theta). \tag{79}$$

To simplify $\boldsymbol{\phi}$, we use Lemma A.1. Denoting $\boldsymbol{u} = (\|\boldsymbol{M}_t^1\|_{\text{nuc}}, \dots, \|\boldsymbol{M}_t^L\|_{\text{nuc}}, \|\boldsymbol{m}_t^\theta\|_1)$, we have

$$\boldsymbol{\phi} = -\text{LMO}_f(\boldsymbol{u}) = -\boldsymbol{D}^{-1}\text{LMO}_\infty(\boldsymbol{D}^{-T}\boldsymbol{u}) = \boldsymbol{D}^{-1}\text{sign}(\boldsymbol{D}^{-T}\boldsymbol{u}) = \boldsymbol{D}^{-1}\mathbf{1}, \tag{80}$$

so that $\phi_\ell = 1$ for $\ell \in [L]$ and $\phi_{L+1} = \lambda = \eta_b/\eta_m$. Plugging back to Equation 79 gives

$$\boldsymbol{W}_{t+1}^\ell = \boldsymbol{W}_t^\ell - \eta_m\text{polar}(\boldsymbol{M}_t^\ell)$$
$$\boldsymbol{\theta}_{t+1} = \boldsymbol{\theta}_t - \eta_a\text{sign}(\boldsymbol{m}_t^\theta), \tag{81}$$

which is exactly the update from Scion (Equation 72). $\qquad\square$

Throughout our paper, $\text{PolarGrad}$ refers to the following algorithm:

$$\boldsymbol{W}_{t+1}^\ell = \boldsymbol{W}_t^\ell - \eta_s\|\boldsymbol{M}_t^\ell\|_{\text{nuc}}\text{polar}(\boldsymbol{M}_t^\ell)$$
$$\boldsymbol{\theta}_{t+1} = \boldsymbol{\theta}_t - \eta_b\frac{\boldsymbol{m}_t^\theta}{\sqrt{\boldsymbol{v}_t^\theta} + \epsilon}. \tag{82}$$

Lau et al. (2025) use the name "PolarGrad" to refer to a class of matrix-aware optimization methods, whereas we use it to refer to the specific method called "Vanilla PolarGrad" by Lau et al. (2025) (see their Equation (8)), with Adam used for non-matrix parameters.

**Proposition A.3.** $\text{PolarGrad}$ is exactly CSD with step size $\eta_m$ with respect to

$$\|\boldsymbol{W}\|_{\text{PG}} = \sqrt{\sum_{\ell=1}^L \|\boldsymbol{W}^\ell\|_{2\to2}^2 + \frac{\eta_m}{\eta_b}\|\boldsymbol{\theta}\|_{\text{ada}2}^2}. \tag{83}$$

*Proof.* Denote $\lambda = \eta_b/\eta_m$. Notice that $\|\cdot\|_{\text{PG}}$ can be written as a composition (as in Lemma 3.3) as:

$$\|\boldsymbol{W}\|_{\text{PG}} = f(g_1(\boldsymbol{W}^1), \dots, g_L(\boldsymbol{W}^L), g_{L+1}(\boldsymbol{\theta})), \tag{84}$$

with $g_i(\boldsymbol{M}) = \|\boldsymbol{M}\|_{2\to2}$ for $i \le L$, $g_{L+1}(\boldsymbol{\theta}) = \|\boldsymbol{\theta}\|_{\text{ada}2}/\sqrt{\lambda}$, and $f(\boldsymbol{v}) = \|\boldsymbol{v}\|_2$. Therefore, $\|\cdot\|_{\text{PG}}$ uses the $\ell_2$ norm as the product norm, so Equation 11 implies that the update can be rewritten as

$$\boldsymbol{W}_{t+1}^\ell = \boldsymbol{W}_t^\ell + \eta_m\|\boldsymbol{M}_t^\ell\|_{\text{nuc}}\text{LMO}_{2\to2}(\boldsymbol{M}_t^\ell)$$
$$\boldsymbol{\theta}_{t+1} = \boldsymbol{\theta}_t + \lambda\eta_m\|\boldsymbol{m}_t^\theta\|_{\text{ada}2,*}\text{LMO}_{\text{ada}2}(\boldsymbol{m}_t^\theta). \tag{85}$$

The update to $\boldsymbol{W}_t^\ell$ can be simplified by plugging in $\text{LMO}_{2\to2}(\boldsymbol{M}) = -\text{polar}(\boldsymbol{M})$, and the update to $\boldsymbol{\theta}_t$ can be simplified by plugging in the definition of $\lambda$ and the dual and LMO of $\|\cdot\|_{\text{ada}2}$ from

Proposition 3.5. This yields that Equation 85 is equivalent to

$$W_{t+1}^{\ell} = W_t^{\ell} - \eta_m \|M_t^{\ell}\|_{\text{nuc}} \text{polar}(M_t^{\ell})$$

$$\boldsymbol{\theta}_{t+1} = \boldsymbol{\theta}_t - \eta_b \frac{\boldsymbol{m}_t^{\theta}}{\sqrt{\boldsymbol{v}_t^{\theta}} + \epsilon}, \tag{86}$$

which is exactly PolarGrad (Equation 82). $\qquad\square$

## B    PROOFS FROM SECTION 4

**Proposition 4.1.** [Constrained Momo] The ball constrained truncated model update is given by

$$\boldsymbol{w}_{t+1} = \underset{\|\boldsymbol{w}-\boldsymbol{w}_t\| \le \eta}{\arg\min} \left\{ \max\left( \tilde{F}_t + \langle \boldsymbol{m}_t, \boldsymbol{w} - \boldsymbol{w}_t \rangle, F_* \right) \right\} \tag{21}$$

$$= \boldsymbol{w}_t + \min\left( \eta, \frac{\tilde{F}_t - F_*}{\|\boldsymbol{m}_t\|_*} \right) \text{LMO}(\boldsymbol{m}_t) \tag{22}$$

*Proof of Proposition 4.1.* Similar to the proofs of Proposition 3.1 and 3.2, we change variables to parameterize the magnitude $r = \|\boldsymbol{w} - \boldsymbol{w}_t\|$ and direction $\boldsymbol{\Delta} = (\boldsymbol{w} - \boldsymbol{w}_t)/\|\boldsymbol{w} - \boldsymbol{w}_t\|$ of the update. So $\boldsymbol{w}_{t+1} = \boldsymbol{w}_t + r_t \boldsymbol{\Delta_t}$, where

$$(r_t, \boldsymbol{\Delta}_t) = \underset{r \in [0,\eta], \|\boldsymbol{\Delta}\|=1}{\arg\min} \left\{ \max\left( \tilde{F}_t + r\langle \boldsymbol{m}_t, \boldsymbol{\Delta} \rangle, F_* \right) \right\}. \tag{87}$$

Since $\max\left( \tilde{F}_t + r\langle \boldsymbol{m}_t, \boldsymbol{\Delta} \rangle, F_* \right)$ is monotonic in $\langle \boldsymbol{m}_t, \boldsymbol{\Delta} \rangle$,

$$\boldsymbol{\Delta}_t = \underset{\|\boldsymbol{\Delta}\|=1}{\arg\min}\langle \boldsymbol{m}_t, \boldsymbol{\Delta} \rangle = \text{LMO}(\boldsymbol{m}_t), \tag{88}$$

so

$$r_t = \underset{r \in [0,\eta]}{\arg\min} \left\{ \max\left( \tilde{F}_t - r\langle \boldsymbol{m}_t, \boldsymbol{\Delta_t} \rangle, F_* \right) \right\} = \underset{r \in [0,\eta]}{\arg\min} \left\{ \max\left( \tilde{F}_t - r\|\boldsymbol{m}_t\|_*, F_* \right) \right\}. \tag{89}$$

Note that $\max\left( \tilde{F}_t - r\|\boldsymbol{m}_t\|_*, F_* \right)$ can have multiple minimizing values of $r \in [0,\eta]$. If $\eta \le (\tilde{F}_t - F_*)/\|\boldsymbol{m}_t\|_*$, then the minimizer $r = \eta$ is unique. If $\eta \ge (\tilde{F}_t - F_*)/\|\boldsymbol{m}_t\|_*$, then any $r$ with $(\tilde{F}_t - F_*)/\|\boldsymbol{m}_t\|_* \le r \le \eta$ achieves the minimum $F_*$. In this case, we choose the value that minimizes the norm of the update, i.e. $r_t = (\tilde{F}_t - F_*)/\|\boldsymbol{m}_t\|_*$. These two cases are summarized as:

$$r_t = \min\left( \eta, \frac{\tilde{F}_t - F_*}{\|\boldsymbol{m}_t\|_*} \right), \tag{90}$$

so

$$\boldsymbol{w}_{t+1} = \boldsymbol{w}_t + \min\left( \eta, \frac{\tilde{F}_t - F_*}{\|\boldsymbol{m}_t\|_*} \right) \text{LMO}(\boldsymbol{m}_t). \tag{91}$$

$\qquad\square$

**Proposition 4.2.** [Regularized Momo] The regularized truncated model update is given by

$$\boldsymbol{w}_{t+1} = \underset{\boldsymbol{w}}{\arg\min} \left\{ \max\left( \tilde{F}_t + \langle \boldsymbol{m}_t, \boldsymbol{w} - \boldsymbol{w}_t \rangle, F_* \right) + \frac{1}{2\eta}\|\boldsymbol{w} - \boldsymbol{w}_t\|^2 \right\} \tag{23}$$

$$= \boldsymbol{w}_t + \min\left( \eta, \frac{\tilde{F}_t - F_*}{\|\boldsymbol{m}_t\|_*^2} \right) \|\boldsymbol{m}_t\|_* \text{LMO}(\boldsymbol{m}_t) \tag{24}$$

*Proof of Proposition 4.2.* Similar to the proofs of Proposition 3.1 and 3.2, we perform a change of variables to parameterize the magnitude $r = \|\boldsymbol{w} - \boldsymbol{w}_t\|$ and direction $\boldsymbol{\Delta} = (\boldsymbol{w} - \boldsymbol{w}_t)/\|\boldsymbol{w} - \boldsymbol{w}_t\|$ of the update. So $\boldsymbol{w}_{t+1} = \boldsymbol{w}_t + r_t \boldsymbol{\Delta}_t$, where

$$(r_t, \boldsymbol{\Delta}_t) = \underset{r \ge 0, \|\boldsymbol{\Delta}\|=1}{\arg\min} \left\{ \max\left( \tilde{F}_t + r\langle \boldsymbol{m}_t, \boldsymbol{\Delta} \rangle, F_* \right) + \frac{r^2}{2\eta} \right\}. \tag{92}$$

Note that $\max\left(\tilde{F}_t + r\langle \boldsymbol{m}_t, \boldsymbol{\Delta}\rangle, F_*\right) + \frac{r^2}{2\eta}$ is monotonic in $\langle \boldsymbol{m}_t, \boldsymbol{\Delta}\rangle$, so

$$\boldsymbol{\Delta}_t = \arg\min_{\|\boldsymbol{\Delta}\|=1}\{\langle \boldsymbol{m}_t, \boldsymbol{\Delta}\rangle\} = \mathsf{LMO}(\boldsymbol{m}_t), \tag{93}$$

and

$$r_t = \arg\min_{r \geq 0}\left\{\max\left(\tilde{F}_t + r\langle \boldsymbol{m}_t, \boldsymbol{\Delta}_t\rangle, F_*\right) + \frac{r^2}{2\eta}\right\} \tag{94}$$

$$= \arg\min_{r \geq 0}\left\{\max\left(\tilde{F}_t - r\|\boldsymbol{m}_t\|_*, F_*\right) + \frac{r^2}{2\eta}\right\}. \tag{95}$$

Denote $f(r) = \max\left(\tilde{F}_t - r\|\boldsymbol{m}_t\|_*, F_*\right) + \frac{r^2}{2\eta}$. Then $f$ can be written piecewise as

$$f(r) = \begin{cases} \tilde{F}_t - r\|\boldsymbol{m}_t\|_* + \frac{r^2}{2\eta} & r \leq \frac{\tilde{F}_t - F_*}{\|\boldsymbol{m}_t\|_*} \\ F_* + \frac{r^2}{2\eta} & r \geq \frac{\tilde{F}_t - F_*}{\|\boldsymbol{m}_t\|_*} \end{cases}. \tag{96}$$

Note that $f$ is increasing for $r \geq (\tilde{F}_t - F_*)/\|\boldsymbol{m}_t\|_*$, so its minimizer is the minimizer of $\tilde{F}_t - r\|\boldsymbol{m}_t\|_* + \frac{r^2}{2\eta}$ for $r \leq (\tilde{F}_t - F_*)/\|\boldsymbol{m}_t\|_*$. So

$$r_t = \min\left(\eta\|\boldsymbol{m}_t\|_*, \frac{\tilde{F}_t - F_*}{\|\boldsymbol{m}_t\|_*}\right), \tag{97}$$

therefore

$$\boldsymbol{w}_{t+1} = \boldsymbol{w}_t + \left(\eta, \frac{\tilde{F}_t - F_*}{\|\boldsymbol{m}_t\|_*^2}\right)\|\boldsymbol{m}_t\|_*\mathsf{LMO}(\boldsymbol{m}_t). \tag{98}$$

Note that this value of $\boldsymbol{w}_{t+1}$ is the unique minimizer of

$$\max\left(\tilde{F}_t + \langle \boldsymbol{m}_t, \boldsymbol{w} - \boldsymbol{w}_t\rangle, F_*\right) + \frac{1}{2\eta}\|\boldsymbol{w} - \boldsymbol{w}_t\|^2, \tag{99}$$

since this function is strongly convex (sum of a convex function and a strongly convex function), and therefore has a unique minimizer. $\qquad\square$

The pseudocode for Constrained Momo and Regularized Momo are shown in Algorithm 2. To see why this algorithm correctly computes $\tilde{F}_t$, note that

$$\tilde{F}_t = \sum_{i=0}^{t} \rho_{t,i}\left(F_i(\boldsymbol{w}_i) + \langle \boldsymbol{g}_i, \boldsymbol{w}_t - \boldsymbol{w}_i\rangle\right) \tag{100}$$

$$= \sum_{i=0}^{t} \rho_{t,i}\left(F_i(\boldsymbol{w}_i) - \langle \boldsymbol{g}_i, \boldsymbol{w}_i\rangle\right) + \sum_{i=0}^{t} \rho_{t,i}\langle \boldsymbol{g}_i, \boldsymbol{w}_t\rangle \tag{101}$$

$$= \sum_{i=0}^{t} \rho_{t,i}\left(F_i(\boldsymbol{w}_i) - \langle \boldsymbol{g}_i, \boldsymbol{w}_i\rangle\right) + \langle \boldsymbol{m}_t, \boldsymbol{w}_t\rangle. \tag{102}$$

So denoting $\tilde{f}_t = \sum_{i=0}^{t} \rho_{t,i}\left(F_i(\boldsymbol{w}_i) - \langle \boldsymbol{g}_i, \boldsymbol{w}_i\rangle\right)$, we have $\tilde{F}_t = \tilde{f}_t + \langle \boldsymbol{m}_t, \boldsymbol{w}_t\rangle$, and

$$\tilde{f}_t = \beta\tilde{f}_{t-1} + (1 - \beta)\left(F_t(\boldsymbol{w}_t) - \langle \boldsymbol{g}_t, \boldsymbol{w}_t\rangle\right), \tag{103}$$

so that $\tilde{f}_t$ is given by the running average used in Algorithm 2.

Now we derive the closed-form update for our proposed algorithm MuonMax-Momo. Algorithm 3 has the pseudocode for the algorithm, and Proposition 4.3 proves that this procedure implements Regularized Momo with respect to $\|\cdot\|_{\mathrm{MM}}$. Note that Algorithm 3 shows the pseudocode with stale nuclear norm approximations, while Proposition 4.3 considers the vanilla version.

It should be noted that, if we set $\beta = 0$, the stepsize scaling $\sum_{\ell=1}^{L} \|\boldsymbol{G}_t^\ell\|_{\mathrm{nuc}}$ for the matrix layers in Algorithm 3 was previously mentioned by Bernstein & Newhouse (2024a) (see their Proposition 5). However, we are not aware of any existing implementation or evaluation of this stepsize scaling, and we found in our experiments that this sort of scaling (without model truncation) is not competitive with Muon.

**Algorithm 2** Momo (Constrained or Regularized)

**Inputs:** $\boldsymbol{w}_0$, learning rate $\eta$, momentum $\beta$, loss lower bound $F_*$

1: **for** $t = 0, 1, \ldots, T-1$ **do**
2:    $\boldsymbol{g}_t \leftarrow \text{backward}(\boldsymbol{w}_t)$
3:    $\boldsymbol{m}_t = \beta \boldsymbol{m}_{t-1} + (1-\beta)\boldsymbol{g}_t$
4:    $\tilde{f}_t = \beta \tilde{f}_{t-1} + (1-\beta)\left(F_t(\boldsymbol{w}_t) - \langle \boldsymbol{g}_t, \boldsymbol{w}_t \rangle\right)$
5:    $\tilde{F}_t = \tilde{f}_t + \langle \boldsymbol{m}_t, \boldsymbol{w}_t \rangle$
6:    **if** Constrained **then**
7:      $\boldsymbol{w}_{t+1} = \boldsymbol{w}_t + \min\left(\eta, \frac{\tilde{F}_t - F_*}{\|\boldsymbol{m}_t\|_*}\right) \text{LMO}(\boldsymbol{m}_t)$
8:    **else**
9:      $\boldsymbol{w}_{t+1} = \boldsymbol{w}_t + \min\left(\eta, \frac{\tilde{F}_t - F_*}{\|\boldsymbol{m}_t\|_*^2}\right) \|\boldsymbol{m}_t\|_* \text{LMO}(\boldsymbol{m}_t)$
10:    **end if**
11: **end for**

**Algorithm 3** MuonMax-Momo

**Inputs:** $\boldsymbol{W}_0^1, \ldots, \boldsymbol{W}_0^L, \boldsymbol{\theta}_0$, learning rates $\eta_m, \eta_b$, EMA parameters $\beta, \beta_2$, loss lower bound $F_*$
**Defaults:** $\eta_m = \eta_b = 0.01$, $\beta = \beta_2 = 0.95$

1: **for** $t = 0, 1, \ldots, T-1$ **do**

  **Update momentum.**
2:    $(\boldsymbol{G}_t^1, \ldots, \boldsymbol{G}_t^L, \boldsymbol{g}_t^\theta) \leftarrow \text{backward}(\boldsymbol{W}_t^1, \ldots \boldsymbol{W}_t^L, \boldsymbol{\theta}_t)$
3:    **for** $\ell = 1, \ldots, L$ **do**
4:      $\boldsymbol{M}_t^\ell = \beta \boldsymbol{M}_{t-1}^\ell + (1-\beta)\boldsymbol{G}_t^\ell$
5:    **end for**
6:    $\boldsymbol{m}_t^\theta = \beta \boldsymbol{m}_{t-1}^\theta + (1-\beta)\boldsymbol{g}_t^\theta$
7:    $\boldsymbol{v}_t^\theta = \beta_2 \boldsymbol{v}_{t-1}^\theta + (1-\beta_2)\boldsymbol{g}_t^\theta \odot \boldsymbol{g}_t^\theta$

  **Update internal truncation variables.**
8:    $\tilde{f}_t = \beta \tilde{f}_{t-1} + (1-\beta)\left(F_t(\boldsymbol{W}_t) - \sum_{\ell=1}^L \langle \boldsymbol{G}_t^\ell, \boldsymbol{W}_t^\ell \rangle - \langle \boldsymbol{g}_t^\theta, \boldsymbol{\theta}_t \rangle\right)$
9:    $\tilde{F}_t = \tilde{f}_t + \sum_{\ell=1}^L \langle \boldsymbol{M}_t^\ell, \boldsymbol{W}_t^\ell \rangle + \langle \boldsymbol{m}_t^\theta, \boldsymbol{\theta}_t \rangle$
10:    $d_t = \sqrt{\left(\sum_{\ell=1}^L d_{t-1}^\ell\right)^2 + \frac{\eta_b}{\eta_m}\left\|\frac{\boldsymbol{m}_t^\theta}{\sqrt{\boldsymbol{v}_t^\theta + \epsilon}}\right\|_2^2}$

  **Update parameters.**
11:    **for** $\ell = 1, \ldots, L$ **do**
12:      $\boldsymbol{P} \leftarrow \text{polar}(\boldsymbol{M}_t^\ell)$
13:      $\boldsymbol{W}_{t+1}^\ell \leftarrow \boldsymbol{W}_t^\ell - \min\left(\eta_m, \frac{\tilde{F}_t - F_*}{d_t^2}\right)\left(\sum_{j=1}^L d_{t-1}^\ell\right)\boldsymbol{P}$
14:      $d_t^\ell \leftarrow \langle \boldsymbol{P}, \boldsymbol{M}_t^\ell \rangle$
15:    **end for**
16:    $\boldsymbol{\theta}_{t+1} = \boldsymbol{\theta}_t - \min\left(\eta_b, \frac{\eta_b}{\eta_m}\frac{\tilde{F}_t - F_*}{d_t^2}\right)\frac{\boldsymbol{m}_t^\theta}{\sqrt{\boldsymbol{v}_t^\theta + \epsilon}}$
17: **end for**

**Proposition 4.3.** [MuonMax-Momo] Regularized Momo with respect to the norm $\|\cdot\|_{\mathrm{MM}}$ as defined in equation 17 has the following closed form:

$$d_t = \sqrt{\left(\sum_{\ell=1}^{L} \|\boldsymbol{M}_t^\ell\|_{\mathrm{nuc}}\right)^2 + \frac{\eta_b}{\eta_m}\left\|\frac{\boldsymbol{m}_t^\theta}{\sqrt{\sqrt{\boldsymbol{v}_t^\theta}+\epsilon}}\right\|_2^2}$$

$$\boldsymbol{W}_{t+1}^\ell = \boldsymbol{W}_t^\ell - \min\left\{\eta_m, \frac{\tilde{F}_t - F_*}{d_t^2}\right\}\left(\sum_{j=1}^{L}\|\boldsymbol{M}_t^j\|_{\mathrm{nuc}}\right)\mathrm{polar}(\boldsymbol{M}_t^\ell) \qquad (25)$$

$$\boldsymbol{\theta}_{t+1} = \boldsymbol{\theta}_t - \min\left\{\eta_b, \frac{\eta_b}{\eta_m}\frac{\tilde{F}_t - F_*}{d_t^2}\right\}\frac{\boldsymbol{m}_t^\theta}{\sqrt{\boldsymbol{v}_t^\theta}+\epsilon}.$$

*Proof of Proposition 4.3.* The proof structure is largely similar to that of Proposition 3.6. By Proposition 4.2, one step of Regularized Momo w.r.t. $\|\cdot\|_{\mathrm{MM}}$ can be written as

$$\boldsymbol{W}_{t+1} = \boldsymbol{W}_t + \min\left(\eta_m, \frac{\tilde{F}_t - F_*}{\|\boldsymbol{M}_t\|_{\mathrm{MM},*}^2}\right)\|\boldsymbol{M}_t\|_{\mathrm{MM},*}\mathsf{LMO}_{\mathrm{MM}}(\boldsymbol{M}_t), \qquad (104)$$

where $\boldsymbol{M}_t$ is the momentum buffer for all network parameters, i.e. it is the concatenation of the momentum buffers of each parameter:

$$\boldsymbol{M}_t = (\boldsymbol{M}_t^1, \ldots, \boldsymbol{M}_t^L, \boldsymbol{m}_t^\theta). \qquad (105)$$

Comparing Equation 104 with Equation 25, we have to prove that $d_t = \|\boldsymbol{M}_t\|_{\mathrm{MM},*}$ and compute $\|\boldsymbol{M}_t\|_{\mathrm{MM},*}\mathsf{LMO}_{\mathrm{MM}}(\boldsymbol{M}_t)$. To do this, we write $\|\cdot\|_{\mathrm{MM}}$ with repeated compositions of norms whose dual and $\mathsf{LMO}$ we already know. Denoting $\lambda = \eta_b/\eta_m$ and

$$f(z_1, z_2) = \sqrt{z_1^2 + \frac{1}{\lambda}z_2^2} \qquad (106)$$

$$g_1(\boldsymbol{W}_1, \ldots, \boldsymbol{W}_L) = \max_{\ell \in [L]}\|\boldsymbol{W}_\ell\|_{2\to 2} \qquad (107)$$

$$g_2(\boldsymbol{\theta}) = \|\boldsymbol{\theta}\|_{\mathrm{ada2}}, \qquad (108)$$

we can write $\|\cdot\|_{\mathrm{MM}}$ as a composition in the notation of Lemma 3.3 as

$$\|\boldsymbol{W}\|_{\mathrm{MM}} = f(g_1(\boldsymbol{W}_1, \ldots, \boldsymbol{W}_L), g_2(\boldsymbol{\theta})). \qquad (109)$$

Further denoting $\boldsymbol{D} = \mathrm{diag}(1, 1/\sqrt{\lambda})$, we can write $f(z_1, z_2) = \|\boldsymbol{D}(z_1, z_2)^T\|_2$. We can now use Lemma 3.3 to compute the dual of $\|\cdot\|_{\mathrm{MM},*}$ as

$$\|\boldsymbol{W}\|_{\mathrm{MM},*} = f_*(g_{1,*}(\boldsymbol{W}_1, \ldots, \boldsymbol{W}_L), g_{2,*}(\boldsymbol{\theta})) \qquad (110)$$

$$\overset{(i)}{=} \sqrt{g_{1,*}^2(\boldsymbol{W}_1, \ldots, \boldsymbol{W}_L) + \lambda g_{2,*}^2(\boldsymbol{\theta})} \qquad (111)$$

$$\overset{(ii)}{=} \sqrt{g_{1,*}^2(\boldsymbol{W}_1, \ldots, \boldsymbol{W}_L) + \lambda\left\|\frac{\boldsymbol{\theta}}{\sqrt{\sqrt{\boldsymbol{v}_t^\theta}+\epsilon}}\right\|^2} \qquad (112)$$

$$\overset{(iii)}{=} \sqrt{\left(\sum_{\ell=1}^{L}\|\boldsymbol{W}_\ell\|_{\mathrm{nuc}}\right)^2 + \lambda\left\|\frac{\boldsymbol{\theta}}{\sqrt{\sqrt{\boldsymbol{v}_t^\theta}+\epsilon}}\right\|^2} \qquad (113)$$

where $(i)$ uses Lemma A.1 to plug in the dual of $f$, $(ii)$ plugs in the dual of $\|\cdot\|_{\mathrm{ada2}}$ which we computed in the proof of Proposition 3.5, and $(iii)$ uses Lemma 3.3 to compute the dual of $g_1$, which itself is a composition $g_1 = \ell_\infty \circ (\|\cdot\|_{2\to 2}, \ldots, \|\cdot\|_{2\to 2})$. This confirms that $d_t = \|\boldsymbol{M}_t\|_{\mathrm{MM},*}$, so

$$\boldsymbol{W}_{t+1} = \boldsymbol{W}_t + \min\left(\eta_m, \frac{\tilde{F}_t - F_*}{d_t^2}\right)d_t\mathsf{LMO}_{\mathrm{MM}}(\boldsymbol{M}_t), \qquad (114)$$

To compute the $\mathsf{LMO}$ of $\|\cdot\|_{\mathrm{MM}}$, we again use Lemma 3.3. Denoting $(\phi_1, \phi_2) = -\mathsf{LMO}_f(g_{1,*}(\boldsymbol{W}_1, \ldots, \boldsymbol{W}_L), g_{2,*}(\boldsymbol{\theta}))$, Lemma 3.3 implies

$$\mathsf{LMO}_{\mathrm{MM}}(\boldsymbol{W}) = (\phi_1\mathsf{LMO}_{g_1}(\boldsymbol{W}_1, \ldots, \boldsymbol{W}_L), \phi_2\mathsf{LMO}_{g_2}(\boldsymbol{\theta})) \qquad (115)$$

$$\overset{(i)}{=} (-\phi_1(\mathrm{polar}(\boldsymbol{W}_1), \ldots, \mathrm{polar}(\boldsymbol{W}_L)), \phi_2\mathsf{LMO}_{g_2}(\boldsymbol{\theta})) \qquad (116)$$

$$\overset{(ii)}{=} -\left(\phi_1(\mathrm{polar}(\boldsymbol{W}_1), \ldots, \mathrm{polar}(\boldsymbol{W}_L)), \phi_2\frac{\boldsymbol{\theta}}{\sqrt{\boldsymbol{v}_t}+\epsilon}\middle/\left\|\frac{\boldsymbol{\theta}}{\sqrt{\sqrt{\boldsymbol{v}_t}+\epsilon}}\right\|_2\right), \qquad (117)$$

where $(i)$ uses Lemma 3.3 to compute the LMO of $g_1$, which again is the composition $g_1 = \ell_\infty \circ (\|\cdot\|_{2\to2}, \ldots, \|\cdot\|_{2\to2})$, and $(iii)$ uses Lemma A.1 to plug in the dual norm of $g_2 = \|\cdot\|_{\text{ada2}}$. The $\phi$ terms can be simplified as

$$(\phi_1, \phi_2) = -\mathsf{LMO}_f(g_{1,*}(\boldsymbol{W}_1, \ldots, \boldsymbol{W}_L), g_{2,*}(\boldsymbol{\theta})) \tag{118}$$

$$\stackrel{(i)}{=} -\mathsf{LMO}_f\left(\sum_{\ell=1}^{L} \|\boldsymbol{W}_\ell\|_{\text{nuc}}, \left\|\frac{\boldsymbol{\theta}}{\sqrt{\sqrt{\boldsymbol{v}_t}+\epsilon}}\right\|_2\right) \tag{119}$$

$$\stackrel{(ii)}{=} -\boldsymbol{D}^{-1}\mathsf{LMO}_2\left(\sum_{\ell=1}^{L} \|\boldsymbol{W}_\ell\|_{\text{nuc}}, \sqrt{\lambda}\left\|\frac{\boldsymbol{\theta}}{\sqrt{\sqrt{\boldsymbol{v}_t}+\epsilon}}\right\|_2\right) \tag{120}$$

$$= \frac{1}{d_t}\boldsymbol{D}^{-1}\left(\sum_{\ell=1}^{L} \|\boldsymbol{W}_\ell\|_{\text{nuc}}, \sqrt{\lambda}\left\|\frac{\boldsymbol{\theta}}{\sqrt{\sqrt{\boldsymbol{v}_t}+\epsilon}}\right\|_2\right) \tag{121}$$

$$= \frac{1}{d_t}\left(\sum_{\ell=1}^{L} \|\boldsymbol{W}_\ell\|_{\text{nuc}}, \lambda\left\|\frac{\boldsymbol{\theta}}{\sqrt{\sqrt{\boldsymbol{v}_t}+\epsilon}}\right\|_2\right), \tag{122}$$

where $(i)$ plugs in the previously computed duals $g_{1,*}$ and $g_{2,*}$, and $(ii)$ uses Lemma A.1 to plug in the LMO of $f$. Plugging the values of $(\phi_1, \phi_2)$ into Equation 117 yields

$$\mathsf{LMO}_{\text{MM}}(\boldsymbol{W}) = -\frac{1}{d_t}\left(\left(\sum_{\ell=1}^{L} \|\boldsymbol{W}_\ell\|_{\text{nuc}}\right)(\text{polar}(\boldsymbol{W}_1), \ldots, \text{polar}(\boldsymbol{W}_L)), \lambda\frac{\boldsymbol{\theta}}{\sqrt{\boldsymbol{v}_t}+\epsilon}\right), \tag{123}$$

and finally, plugging this back into Equation 114 yields

$$\boldsymbol{W}_{t+1}^\ell = \boldsymbol{W}_t - \min\left(\eta_m, \frac{\tilde{F}_t - F_*}{d_t^2}\right)\left(\sum_{i=1}^{L} \|\boldsymbol{W}_i\|_{\text{nuc}}\right)\text{polar}(\boldsymbol{M}_t^\ell) \tag{124}$$

$$\boldsymbol{\theta}_{t+1} = \boldsymbol{\theta}_t - \min\left(\eta_m, \frac{\tilde{F}_t - F_*}{d_t^2}\right)\lambda\frac{\boldsymbol{m}_t^\theta}{\sqrt{\boldsymbol{v}_t^\theta}+\epsilon} = \boldsymbol{\theta}_t - \min\left(\eta_b, \frac{\eta_b}{\eta_m}\frac{\tilde{F}_t - F_*}{d_t^2}\right)\frac{\boldsymbol{m}_t^\theta}{\sqrt{\boldsymbol{v}_t^\theta}+\epsilon}, \tag{125}$$

which is exactly the update in Equation 25. □

## C  EXPERIMENTAL DETAILS

**Setup**   We did not use weight decay or Nesterov momentum, as we found both to have very small effects on final loss. All methods use a warmup-stable-decay learning rate schedule, where the learning rate is linearly warmed up for the first 5% of steps, held constant until halfway through training, then linearly decayed to 10% of the warmed up value. We use a context length of 1024 and a batch size of 512. Rather than the Newton-Schulz iterations of the original Muon implementation, we use the PolarExpress algorithm (Amsel et al., 2025) to compute approximate polar factors. In this implementation, the weights and gradients are computed in `float32`, whereas the polar factor is computed in `bfloat16` by the PolarExpress (Amsel et al., 2025).

**Tuning Protocol**   For the experiments with FineWeb data in Section 5.1, we tune 36 variations of steepest descent using an iterated grid search to for the two learning rates $\eta_m$ and $\eta_b$. For the 18 variations without model truncation, we first fix the base learning rate at an intermediate value $\eta_b =$1e-3, then tune the Muon learning rate with grid search over $\eta_m \in \{$1e-3, 1e-2, 1e-1, 1$\}$. Some algorithms diverged with $\eta_b =$1e-3, and for these algorithms we instead used $\eta_b =$1e-6 and searched over $\eta_m \in \{$1e-6, 1e-5, 1e-4, 1e-3$\}$. For those algorithms that used $\eta_b =$1e-6 for the first phase, we instead search over $\eta_b \in \{$1e-7, 1e-6, 1e-5, 1e-4$\}$ in the second phase. Finally, for all of these grid searches, we extend the search space individually for each algorithm until the best LR is not a boundary point of the search space. The resulting tuned LRs are shown in Table 2.

For the 18 variations with model truncation, rather than entirely retuning all algorithms, we reuse the tuned LR ratio $\eta_m/\eta_b$ and do a single grid search where $\eta_m$ and $\eta_b$ scale together. More specifically, we run each algorithm with LRs $(\rho\eta_m, \rho\eta_b)$, where $(\eta_m, \eta_b)$ are the LRs tuned for each algorithm without truncation, and the scaling factor $\rho$ ranges over $\rho \in \{0.3, 1, 3, 10, 30, 100\}$. We found that

Table 2: Final validation losses for all variations without model truncation.

| (SD type, Product Norm, Backup Norm) | Muon LR | Other LR | Final Loss | Name |
|---|---|---|---|---|
| (Regularized, $\|\cdot\|_\infty, \|\cdot\|_\infty$) | 1e-3 | 1e-5 | 3.783 | - |
| (Constrained, $\|\cdot\|_\infty, \|\cdot\|_\infty$) | 1e-2 | 1e-3 | 3.599 | Scion |
| (Regularized, $\|\cdot\|_2, \|\cdot\|_\infty$) | 1e-1 | 1e-6 | 4.179 | - |
| (Constrained, $\|\cdot\|_2, \|\cdot\|_\infty$) | 1e-1 | 1e-2 | 3.712 | - |
| (Regularized, $\|\cdot\|_{\text{hyb}}, \|\cdot\|_\infty$) | 1e-3 | 1e-5 | 3.826 | - |
| (Constrained, $\|\cdot\|_{\text{hyb}}, \|\cdot\|_\infty$) | 1e-2 | 1e-3 | 3.610 | - |
| (Regularized, $\|\cdot\|_\infty, \|\cdot\|_{\text{ada}\infty}$) | 1e-3 | 1e-5 | 3.859 | - |
| (Constrained, $\|\cdot\|_\infty, \|\cdot\|_{\text{ada}\infty}$) | 1e-2 | 1e-3 | 3.604 | Muon |
| (Regularized, $\|\cdot\|_2, \|\cdot\|_{\text{ada}\infty}$) | 1e-1 | 1e-4 | 4.229 | - |
| (Constrained, $\|\cdot\|_2, \|\cdot\|_{\text{ada}\infty}$) | 1e-1 | 1e-2 | 3.748 | - |
| (Regularized, $\|\cdot\|_{\text{hyb}}, \|\cdot\|_{\text{ada}\infty}$) | 1e-3 | 1e-4 | 3.917 | - |
| (Constrained, $\|\cdot\|_{\text{hyb}}, \|\cdot\|_{\text{ada}\infty}$) | 1e-2 | 1e-2 | 3.628 | - |
| (Regularized, $\|\cdot\|_\infty, \|\cdot\|_{\text{ada2}}$) | 1e-3 | 1e-4 | 3.757 | - |
| (Constrained, $\|\cdot\|_\infty, \|\cdot\|_{\text{ada2}}$) | 1e-2 | 1e-3 | 3.701 | - |
| (Regularized, $\|\cdot\|_2, \|\cdot\|_{\text{ada2}}$) | 1e-1 | 1e-3 | 4.049 | PolarGrad |
| (Constrained, $\|\cdot\|_2, \|\cdot\|_{\text{ada2}}$) | 1e-1 | 1e-2 | 3.664 | - |
| (Regularized, $\|\cdot\|_{\text{hyb}}, \|\cdot\|_{\text{ada2}}$) | 1e-3 | 1e-3 | 3.791 | MuonMax |
| (Constrained, $\|\cdot\|_{\text{hyb}}, \|\cdot\|_{\text{ada2}}$) | 1e-2 | 1e-2 | 3.585 | - |

the best value of $\rho$ for each algorithm was always at least 1 and at most 30. The resulting tuned LRs are shown in Table 3.

**Hybrid Norm Definition**  Recall that Muon-Max is defined as regularized steepest descent with respect to the following norm:

$$\|\boldsymbol{W}\|_{\text{MM}} = \sqrt{\left(\max_{\ell\in[L]}\|\boldsymbol{W}^\ell\|_{2\to2}\right)^2 + \frac{\eta_m}{\eta_b}\|\boldsymbol{\theta}\|_{\text{ada2}}^2}. \tag{126}$$

This norm fits into our framework by assigning the spectral norm to each weight matrix $\boldsymbol{W}_\ell$, assigning $\|\cdot\|_{\text{ada2}}$ to the remaining parameters, and aggregating norms for all parameters with the following "hybrid" product norm:

$$\|(v_1,\ldots,v_L,v_{L+1})\|_{\text{hyb}} = \sqrt{\left(\max_{\ell\in[L]}v_\ell\right)^2 + \frac{\eta_m}{\eta_b}v_{L+1}^2}. \tag{127}$$

# D  ADDITIONAL EXPERIMENTAL RESULTS

## D.1  FINEWEB

The final validation loss reached by all 36 of our evaluated methods is shown in Tables 2 and 3. Each method is denoted as a 3-tuple of settings from our steepest descent framework: regularized vs constrained steepest descent, product norm, and norm for parameters besides hidden weight matrices.

For the methods without model truncation (Table 2), we see that the RSD methods struggle generally lag behind the CSD methods, likely due to a lack of update normalization. For the CSD methods, Muon and Scion are among the best variations, though the best performing method is actually (Constrained, $\|\cdot\|_{\text{hyb}}, \|\cdot\|_{\text{ada2}}$) (we will return to discuss this method shortly).

For the methods with model truncation (Table 3), we see that both CSD and RSD methods are competitive, meaning that in general model truncation helped RSD methods more than CSD methods (at least in terms of final loss with tuned LRs). Muon-Momo has the lowest loss at 3.551 and Scion-Momo is again among the best performers, but actually many methods achieve losses very close to 3.551. Again, we see that (Constrained, $\|\cdot\|_{\text{hyb}}, \|\cdot\|_{\text{ada2}}$) achieves a very low loss, only being outperformed by Muon-Momo.

The method (Constrained, $\|\cdot\|_{\text{hyb}}, \|\cdot\|_{\text{ada2}}$) is quite similar to our proposed method Muon-Max, the only difference being the use of a normalized update. While this method does achieve a lower

Table 3: Final validation losses for all variations with model truncation.

| (SD type, Product Norm, Backup Norm) | Muon LR | Other LR | Final Loss | Name |
|---|---|---|---|---|
| (Regularized, $\|\cdot\|_\infty$, $\|\cdot\|_\infty$) | 1e-2 | 1e-4 | 3.627 | - |
| (Constrained, $\|\cdot\|_\infty$, $\|\cdot\|_\infty$) | 1e-2 | 1e-3 | 3.592 | Scion-Momo |
| (Regularized, $\|\cdot\|_2$, $\|\cdot\|_\infty$) | 1 | 1e-5 | 3.728 | - |
| (Constrained, $\|\cdot\|_2$, $\|\cdot\|_\infty$) | 1e-1 | 1e-2 | 3.843 | - |
| (Regularized, $\|\cdot\|_{hyb}$, $\|\cdot\|_\infty$) | 1e-2 | 1e-4 | 3.628 | - |
| (Constrained, $\|\cdot\|_{hyb}$, $\|\cdot\|_\infty$) | 3e-2 | 3e-3 | 3.604 | - |
| (Regularized, $\|\cdot\|_\infty$, $\|\cdot\|_{ada\infty}$) | 3e-2 | 3e-4 | 3.578 | - |
| (Constrained, $\|\cdot\|_\infty$, $\|\cdot\|_{ada\infty}$) | 3e-2 | 3e-3 | 3.551 | Muon-Momo |
| (Regularized, $\|\cdot\|_2$, $\|\cdot\|_{ada\infty}$) | 1 | 1e-3 | 3.719 | - |
| (Constrained, $\|\cdot\|_2$, $\|\cdot\|_{ada\infty}$) | 1e-1 | 1e-2 | 3.737 | - |
| (Regularized, $\|\cdot\|_{hyb}$, $\|\cdot\|_{ada\infty}$) | 3e-2 | 3e-3 | 3.584 | - |
| (Constrained, $\|\cdot\|_{hyb}$, $\|\cdot\|_{ada\infty}$) | 3e-2 | 3e-2 | 3.607 | - |
| (Regularized, $\|\cdot\|_\infty$, $\|\cdot\|_{ada2}$) | 3e-3 | 3e-4 | 3.662 | - |
| (Constrained, $\|\cdot\|_\infty$, $\|\cdot\|_{ada2}$) | 1e-2 | 1e-3 | 3.701 | - |
| (Regularized, $\|\cdot\|_2$, $\|\cdot\|_{ada2}$) | 3 | 3e-2 | 3.613 | PolarGrad-Momo |
| (Constrained, $\|\cdot\|_2$, $\|\cdot\|_{ada2}$) | 3e-1 | 3e-2 | 3.602 | - |
| (Regularized, $\|\cdot\|_{hyb}$, $\|\cdot\|_{ada2}$) | 1e-2 | 1e-2 | 3.576 | MuonMax-Momo |
| (Constrained, $\|\cdot\|_{hyb}$, $\|\cdot\|_{ada2}$) | 3e-2 | 3e-2 | 3.553 | - |

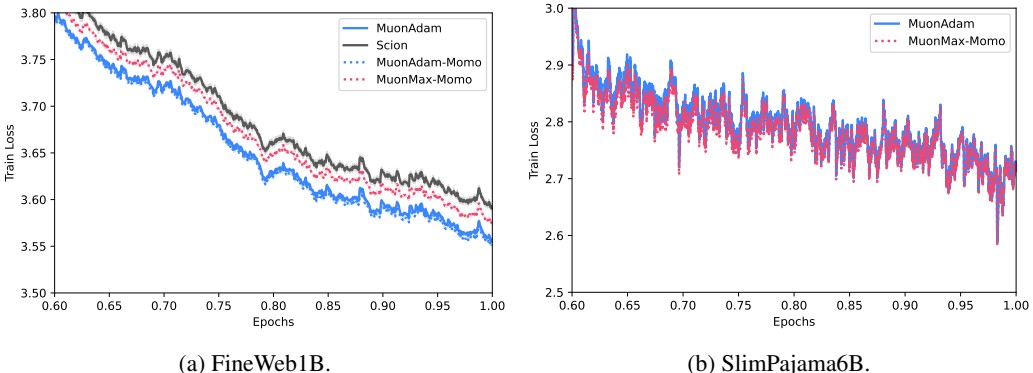

(a) FineWeb1B.  (b) SlimPajama6B.

Figure 4: Training loss for the last 40% of training for FineWeb1B (left) and SlimPajama6B (right).

loss after tuning than MuonMax, we found that this method was not as robust to learning rate tuning. So this method was bested by MuonAdam-Momo in terms of final loss, and it was bested by MuonMax-Momo in terms of learning rate sensitivity, and for this reason we did not perform further evaluations with this method.

We include loss curves for the last 40% of training for the best variations (with tuned learning rates) in Figure 4a, and the final losses reached by the best variations (over three seeds) in Table 4. Also, Figure 5 shows a comparison of MuonAdam, Scion, MuonMax against their truncated counterparts.

Table 4: Validation loss for FineWeb1B with tuned LRs (mean $\pm$ std over three seeds).

| MuonAdam | Scion | MuonAdam-Momo | MuonMax-Momo |
|---|---|---|---|
| $3.5592 \pm 0.0014$ | $3.5947 \pm 0.0031$ | $3.5546 \pm 0.0004$ | $3.5779 \pm 0.0007$ |

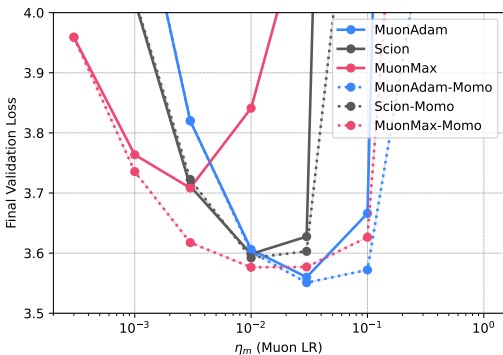

Figure 5: Effect of model truncation on final validation loss. Note that for these runs, we did not use stale nuclear norm approximations in order to isolate the effect of model truncation.

Table 5: Hyperparameter information for additional baselines. For all baselines, we tune the learning rate over $10^{-5+i/4}$ for $i \in \{0, \ldots, 15\}$ once with weight decay on, and again with weight decay off. For each baseline, we set the weight decay coefficient $\lambda$ according to the default value for that algorithm, listed below.

| | Tuned LR ($\lambda = 0$) | Tuned LR ($\lambda > 0$) | Alg-specific parameters |
|---|---|---|---|
| AdamW | $\eta = $ 1e-3 | $\eta = $ 1e-3 ($\lambda = 0.01$) | $\beta_1 = 0.95, \beta_2 = 0.95$ |
| Lion | $\eta = $ 5.6e-5 | $\eta = $ 5.6e-5 ($\lambda = 0.01$) | $\beta_1 = 0.95, \beta_2 = 0.98$ |
| Adan | $\eta = $ 1.8e-3 | $\eta = $ 1e-3 ($\lambda = 0.02$) | $\beta_1 = 0.02, \beta_2 = 0.08, \beta_3 = 0.01$ |
| Sophia | $\eta = $ 5.6e-5 | $\eta = $ 5.6e-5 ($\lambda = 0.2$) | $\beta_1 = 0.965, \beta_2 = 0.99, \rho = 0.05,$ Hessian estimator: GNB |

### D.1.1 ADDITIONAL BASELINES

Here we evaluate four additional baselines (AdamW, Lion, Adan, Sophia) for the GPT2-Small/FineWeb1B setting. We use the same hyperparameters as outlined in Section C, such as batch size, learning rate schedule, random seeds, etc. For a fair comparison with the Muon-type algorithms from the main paper, we allow the same computational budget for hyperparameter tuning. In particular, since the Muon-type algorithms have two learning rates that were tuned over four possible values each, we tune the learning rate of our additional baselines over 16 values. We also evaluate each additional base both with and without weight decay. For weight decay coefficients and algorithm-specific hyperparameters, we use each baseline's default settings as reported in their respective papers. The complete search range and tuned values for each algorithm are shown in Table 5.

The losses for each additional baseline are shown in Figure 6. The most important feature to notice is that none of these baselines reach as low a loss as the baselines from the main paper: from Figure 2a, all of MuonAdam, Scion, MuonAdam-Momo, and MuonMax-Momo achieve less than 3.6 loss with a tuned LR and less than 3.75 loss with multiple LRs in the grid. All of the additional baselines except for AdamW never get below 4.0 validation loss, and AdamW at its best reaches only 3.759. Note also that each of the four baselines were run with and without weight decay, which did not make a significant difference. This means that each baseline was allowed a total of 32 hyperparameter configurations, which is significantly more than the Muon-type algorithms from Figure 2a. We conclude that AdamW is the only additional baseline that is competitive with the aforementioned Muon-type algorithms, though it is still decisively outperformed.

### D.1.2 ADDITIONAL METRICS

We further quantify the efficiency and learning rate sensitivity of each method in Table 6. Efficiency is measured in terms of token throughput, time per training step, time and tokens to reach a target loss, and the "width" of the basin of the LR sensitivity curves. We define the LR basin width as

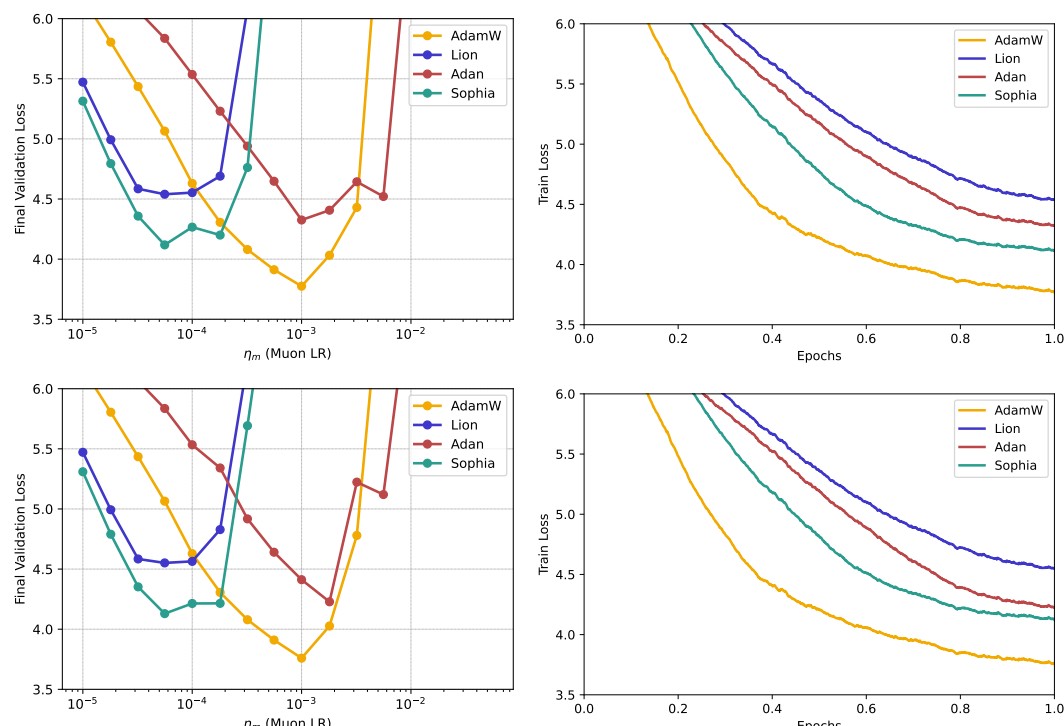

Figure 6: Loss for additional baselines when training GPT2-Small on FineWeb1B. Top row shows results with weight decay, bottom row without weight decay. Left: Validation loss for a sweep over learning rates. Right: Training loss curves for each baseline with tuned learning rate.

follows: for a learning rate sweep of a particular method, let $\ell$ be the loss achieved by the best learning rate, and let $\eta_0$ (respectively $\eta_1$) be the smallest (respectively largest) learning rate in the grid that achieves less than $1.2\ell$ loss. The LR basin width is then defined as $\log_{10}(\eta_1/\eta_0)$. Essentially, the basin width quantifies the orders of magnitude by which the learning rate can vary while still achieving a reasonable loss of 1.2 times the optimum. For throughput and time per step, we average over all learning rates, while for time-to-target, tokens-to-target, and perplexity we choose the best value over all learning rates. Similar metrics for our GPT2-Large/SlimPajama experiments are given in Tables 7 and 8 in Section D.2.1.

From Tables 6, 7, 8, we see that the best perplexity is always achieved by either MuonAdam-Momo or MuonMax-Momo, and these two algorithms significantly outperform all baselines in terms of learning rate sensitivity as quantified by the LR basin width. Compared to MuonAdam, our two proposed algorithms take about 5% more time per step for GPT2-Small and 1.5% to 3% more for GPT2-Large. Also, with tuned LRs, our two proposed algorithms reach the target loss with as many or fewer tokens than all other baselines. The time to target loss of our algorithms with tuned LRs is sometimes better, sometimes worse than MuonAdam, but the gap is never more than 5% for MuonMax-Momo and 10% for MuonAdam-Momo. These results reinforce our finding that our proposed algorithms are significantly more robust to learning rate tuning than baselines, with similar or improved performance after tuning and only a modest increase in wall-clock time.

### D.1.3 VARYING BATCH SIZE

In this section, we perform learning rate sweeps for MuonAdam, Scion, MuonAdam-Momo, and MuonMax-Momo with varying batch sizes. We evaluate these algorithms on the GPT2-Small/FineWeb1B setup detailed in Sections 5.1 and C. Previously we used a batch size of 512; in this section, we vary the batch size over $\{128, 256, 512, 1024, 2048\}$ while keeping all other settings the same. Note that we reuse the tuned LR ratio $\eta_m/\eta_b$ as detailed in Section C.

Table 6: Additional metrics for GPT2-Small/FineWeb1B, including the additional baselines from Section D.1.1. The target loss for this setting is 3.8, which we chose to be small enough to discriminate between the best methods and large enough to include AdamW.

| | Best Perplexity | Throughput (tok/s) | Time per step (s) | Time to target loss (s) | Tokens to target loss | LR Basin W idth |
|---|---|---|---|---|---|---|
| MuonAdam | 33.21 | 938K | 0.559 | **503.31** | **458M** | 1.523 |
| Scion | 34.40 | 948K | 0.554 | 631.51 | 548M | 1.477 |
| MuonAdam-Momo | **33.14** | 894K | 0.587 | 566.82 | **458M** | 2.523 |
| MuonMax-Momo | 33.95 | 892K | 0.588 | 530.04 | **458M** | **3.523** |
| AdamW | 40.99 | 990K | 0.530 | 840.61 | 787M | 1.000 |
| Lion | 92.11 | **996K** | **0.527** | - | - | 1.000 |
| Adan | 66.46 | 957K | 0.548 | - | - | 0.750 |
| Sophia | 59.48 | 974K | 0.539 | - | - | 1.250 |

For each batch size, the final validation loss reached by each algorithm with varying learning rate is shown in Figure 7. Overall, we find the results with different batch size to be largely consistent with our previous results. For all batch sizes, both MuonAdam-Momo and MuonMax-Momo have a wider range of competitive learning rates than MuonAdam and Scion, and for all batch sizes greater than 512, both MuonAdam-Momo and MuonMax-Momo outperform the baselines for every learning rate we tried.

### D.1.4 QWEN2MOE MODEL

We also compared the best performing methods MuonAdam, Scion, MuonMax-Momo and MuonAdam-Momo, when training a Mixture of Experts type model. We trained two Qwen2-MoE variants on the FineWeb dataset. The small variant has hidden size 256, 12 decoder layers, 4 attention heads with 64-d head size, 8 experts per layer, top–2 routing, per-expert FFN width 1280, and shared expert width 1536. The medium variant has hidden size 768, 16 decoder layers, 12 attention heads with 64-d head size, 4 experts per layer, top-2 routing, per-expert FFN width 4096, and shared expert width 4096. Both variants use tied embeddings, context length 1024, and micro-batch size 32 (global batch size 512 sequences).

These configurations emphasize MoE capacity while keeping active parameters per token modest for rapid ablation, following prior sparse expert designs (Shazeer et al., 2017; Fedus et al., 2021; Du et al., 2022). Qwen2-MoE architectural choices and implementation details are taken from the official Qwen2 repository (Team, 2024). No intermediate validation was performed (only end-of-epoch loss) to minimize overhead.

For the two Momo methods we use $F^* = 3.2$ without tuning or trying other values. This turned out to be far from the loss achieved by this small MoE model.

The final loss for each algorithm across a range of learning rates is shown in Figure 8. Indeed, again we found MuonMax-Momo to be the most stable method; for the small variant, MuonMax-Momo achieves the lowest loss for every learning rate we tried. The best loss over all learning rates achieved by each method is given in the following table.

| | Qwen2-MoE Small | Qwen2-MoE Medium |
|---|---|---|
| MuonAdam | 5.2072 | 4.8995 |
| Scion | 5.2156 | 4.7504 |
| MuonAdam-Momo | 5.1975 | 4.8184 |
| MuonMax-Momo | 5.2017 | 4.6561 |

### D.2 SLIMPAJAMA

Figure 9 shows a 2D visualization of final validation losses for Muon, Scion, MuonAdam-Momo, and MuonMax-Momo as the two learning rates vary. We find MuonMax-Momo to be most stable to changes in the learning rates, with both Muon and Scion suffering high losses when the base LR

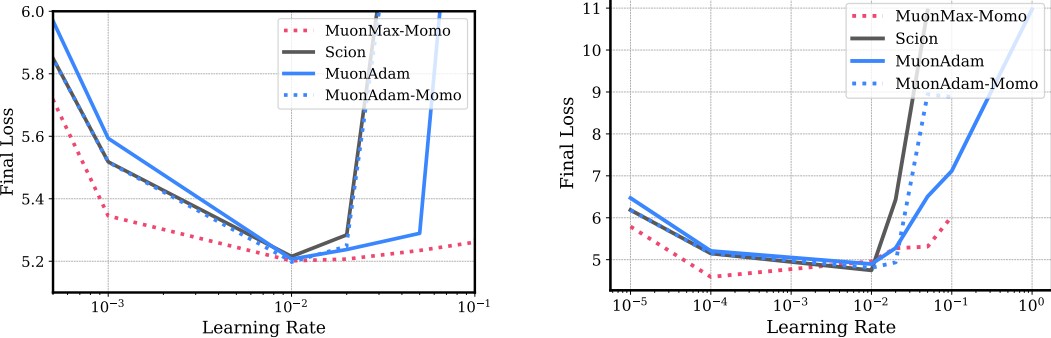

(a) Batch size 128.

(b) Batch size 256.

(c) Batch size 512.

(d) Batch size 1024.

(e) Batch size 2048.

Figure 7: Learning rate sensitivity for GPT2-Small/FineWeb1B with varying batch sizes.

Figure 8: Final loss of Qwen2-MoE for FineWeb1B. Left: Small architecture with 1B tokens. Right: Medium architecture with 10B tokens.

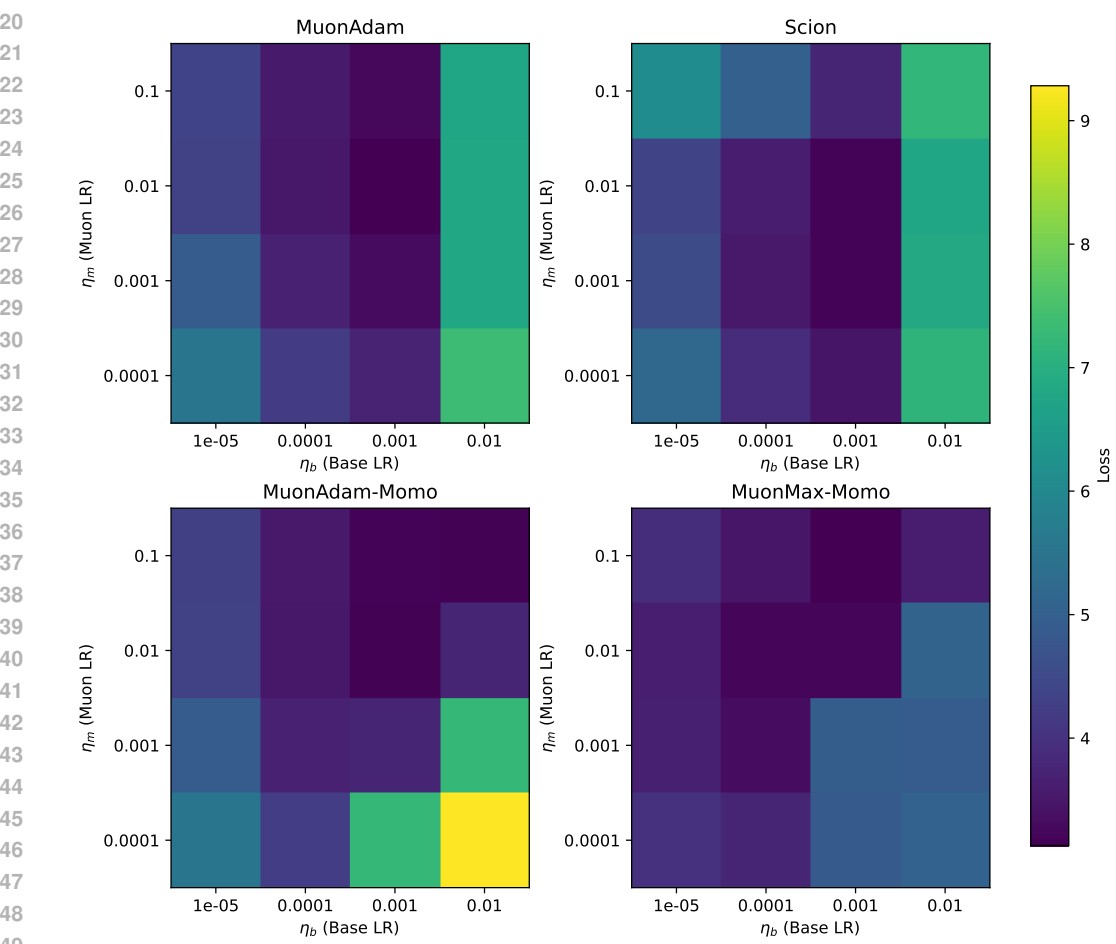

Figure 9: 2D learning rate sensitivity for SlimPajama1B.

Table 7: Additional metrics for GPT2-Large/SlimPajama1B. The target loss for this setting is 3.3.

|  | Best Perplexity | Throughput (tok/s) | Time per step (s) | Time to target loss (s) | Tokens to target loss | LR Basin W idth |
|---|---|---|---|---|---|---|
| MuonAdam | 25.40 | 126K | 4.138 | 3172.77 | 389M | 1.000 |
| Scion | 26.69 | **127K** | **4.114** | 3164.58 | 389M | 1.000 |
| MuonAdam-Momo | 25.51 | 125K | 4.164 | **2237.39** | **275M** | 4.000 |
| MuonMax-Momo | **24.84** | 125K | 4.178 | 2309.51 | **275M** | **5.000** |

$\eta_b$ is large. Interestingly, Muon-Momo has the highest loss when the Muon LR $\eta_m$ is small and the base LR $\eta_b$ is large.

We also include loss curves for the last 40% of training for MuonAdam and MuonMax-Momo (with tuned learning rates) in Figure 4b.

### D.2.1 ADDITIONAL METRICS

Similarly to Section D.1.2, here we include additional metrics to quantify efficiency and LR robustness for our SlimPajama experiments. The definition of each metric is the same as Section D.1.2, here we only change the target loss to 3.3 for GPT2-Large/SlimPajama1B and 3.0 for GPT2-Large/SlimPajama6B. The results are shown in Tables 7 and 8, and are largely similar to those of Table 6. See Section D.1.2 for a discussion of Tables 6, 7, and 8 together.

Table 8: Additional metrics for GPT2-Large/SlimPajama6B. The target loss for this setting is 3.0.

|  | Best Perplexity | Throughput (tok/s) | Time per step (s) | Time to target loss (s) | Tokens to target loss | LR Basin W idth |
|---|---|---|---|---|---|---|
| MuonAdam | 17.29 | **127K** | **4.124** | 4309.90 | **531M** | 1.000 |
| MuonMax-Momo | **17.21** | 123K | 4.239 | **4308.02** | **531M** | **2.000** |

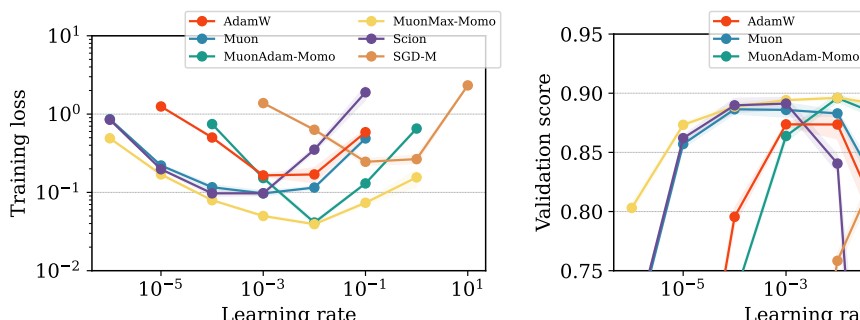

Figure 10: CIFAR-10 with ResNet20. Note that the x-axis shows the base learning rate $\eta_b$, while the Muon learning rate $\eta_m$ is set according to the tuned ratio $\eta_m/\eta_b$.

### D.3 IMAGE CLASSIFICATION

In order to benchmark our proposed algorithms in a variety of settings, here we evaluate our proposed algorithms and baselines for image classification tasks. We evaluate SGD-M (SGD with momentum), AdamW, MuonAdam, Scion, MuonAdam-Momo, and MuonMax-Momo, first for training a ResNet20 (He et al., 2016) on CIFAR-10, then for a ResNet110 on CIFAR-100.

**Setup** All methods use a batch size of 128 and cross entropy loss. We train for 50 epochs with ResNet20/CIFAR-10 and 100 epochs with ResNet110/CIFAR-100. We use a constant learning rate schedule and we do not use weight decay. For SGD-M we set the momentum to 0.9. For AdamW, MuonAdam, MuonAdam-Momo, and MuonMax-Momo, we set $\beta_1 = \beta_2 = 0.95$. We use standard data augmentation for CIFAR: normalization, random horizontal flipping, and random cropping. For Muon-type algorithms, we assign the spectral norm to weights of convolutional layers except for the first convolution in the network. We interpret each convolutional filter as a matrix by flattening all dimensions after the first two, following an early implementation of Muon for image classification (Jordan, 2024).

For the algorithms with two learning rates, i.e. the Muon-type algorithms, we tune the ratio between learning rates with a double grid search. The base LR $\eta_b$ is tuned over $\{1e\text{-}5, 1e\text{-}4, 1e\text{-}3, 1e\text{-}2\}$, and the Muon LR $\eta_m$ is tuned over $\{1e\text{-}4, 1e\text{-}3, 1e\text{-}2, 1e\text{-}1\}$. To benchmark the learning rate sensitivity with a 1D sweep, we then fix the ratio $\eta_m/\eta_b$ and vary $\eta_b$ over $\{1e\text{-}6, 1e\text{-}5, 1e\text{-}4, 1e\text{-}3, 1e\text{-}2, 1e\text{-}1, 1\}$. For SGD-M and AdamW, we sweep the learning rate over $\{1e\text{-}3, 1e\text{-}2, 1e\text{-}1, 1, 10\}$ and $\{1e\text{-}5, 1e\text{-}4, 1e\text{-}3, 1e\text{-}2, 1e\text{-}1\}$, respectively. For the final sweep, we run three random seeds for each algorithm/LR.

**Results** The final training losses and validation accuracies for CIFAR-10 and CIFAR-100 are shown in Figures 10 and 11, respectively. The plots show the mean plus/minus one standard deviation across three seeds.

For CIFAR-10, MuonMax-Momo achieves the lowest training loss for every learning rate and the highest validation accuracy for nearly every learning rate. Here, MuonMax-Momo has achieved the best of both worlds: it reaches the strongest performance with a tuned learning rate and the

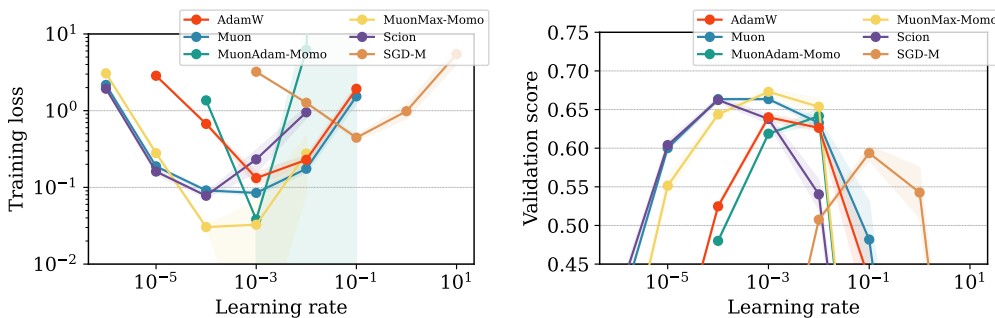

Figure 11: CIFAR-100 with ResNet110. Note that the x-axis shows the base learning rate $\eta_b$, while the Muon learning rate $\eta_m$ is set according to the tuned ratio $\eta_m/\eta_b$.

widest basin across learning rates. Interestingly, while MuonAdam-Momo achieves nearly the best training loss for a tuned learning rate, its basin is much thinner than that of MuonMax-Momo. MuonAdam and Scion are competitive in terms of validation accuracy, while SGD-M and AdamW lag behind in terms of both training loss and validation accuracy.

For CIFAR-100, all algorithms appear more sensitive to the choice of learning rate, and MuonMax-Momo again achieves the lowest loss and highest validation accuracy with a tuned learning rate. Again, MuonAdam-Momo is more sensitive than MuonMax-Momo, and the ranking of the remaining baselines is similar to CIFAR-10.

