# OpenReview forum: "An Exploration of Non-Euclidean Gradient Descent: Muon and its Many Variants"
_ICLR.cc/2026/Conference — Submitted to ICLR 2026_

### Official Review · Reviewer_27dR · 2025-10-27

**Soundness:** 3
**Presentation:** 3
**Contribution:** 2
**Rating:** 6
**Confidence:** 3

**Summary:**

The paper formalizes Muon/Scion-style optimizers as steepest descent under non-Euclidean product norms, showing that practical MuonAdam is exactly constrained steepest descent w.r.t. a norm that aggregates spectral norms with an adaptive infinite norm. It then (i) generalizes Momo to arbitrary norms; (ii) proposes MuonMax, together with a stale dual-norm approximation that avoids extra polar computations; and (iii) provides controlled experiments on GPT-2-scale LMs over FineWeb/SlimPajama, showing substantially improved LR robustness with near-Muon’s memory and small wall-clock overhead.

**Strengths:**

1. Unifying theory with practical implications. Clear derivations for constrained/regularized steepest descent, LMO/dual norms on product spaces, and an exact reinterpretation of Adam as steepest descent under adaptive norms—bridging disparate practices into one framework.
2. Actionable new variants. MuonMax emerges naturally from the framework and, when combined with Momo, exhibits wide learning-rate basins and lower seed variance; the stale nuclear-norm trick is elegant and empirically effective.
3. Clarity and completeness. Propositions/Lemmas are precise; pseudocode and ablations aid reproducibility. The study isolates design axes and reports across them.
4. Relevance. Addresses an active area and offers a principled recipe to reduce hyper-parameter tuning burden in new tasks.

**Weaknesses:**

1. 	Scale & metrics. Experiments stop at GPT-2-Large on next-token loss; no downstream evaluations (e.g., perplexity-to-task correlation, zero-shot LM benchmarks) or larger-scale LLMs to confirm external validity.
2. Theory for “stale” approximation. The stale dual-norm update is well motivated but lacks stability/error bounds; only empirical justification is provided
3. Ablation breadth. Useful ablations are present, yet a deeper comparison against other non-Euclidean/product-norm choices (e.g., modular norm training) and recent LMO variants would strengthen claims of generality

**Questions:**

Please see weaknesses

---

> ### Author Response · Authors · 2025-11-25
>
> Thank you for your review. Below we have responded individually to your points.
>
> **Q1**: Scale & metrics. Experiments stop at GPT-2-Large on next-token loss; no downstream evaluations (e.g., perplexity-to-task correlation, zero-shot LM benchmarks) or larger-scale LLMs to confirm external validity.
>
> **A1**: Due to computational cost we are not able to run larger-scale LLMs, however we have expanded the scope of our experiments by adding new baselines, an MoE archicture, and image classification experiments with ResNets. The results confirm the tuning robustness of our algorithms in a more diverse range of settings than we had initially included. Please see our general response for a summary of these new experiments.
>
> **Q2**: Theory for “stale” approximation. The stale dual-norm update is well motivated but lacks stability/error bounds; only empirical justification is provided.
>
> **A2**: Thanks for the suggestion. The error in the stale approximation could potentially be bounded by standard assumptions like smoothness and small learning rates, however we feel that such an analysis would not say anything convincing about practical setups with real neural networks that don't satisfy such assumptions. We defer instead to the experimental results, which clearly show that the stale update is a very close approximation.
>
> **Q3**: Ablation breadth. Useful ablations are present, yet a deeper comparison against other non-Euclidean/product-norm choices (e.g., modular norm training) and recent LMO variants would strengthen claims of generality.
>
> **A3**: Thanks for the suggestion. Did you have any specific algorithms in mind besides the modular norm method? As we discussed in our related work, the modular norm paper (Bernstein & Newhouse, 2024b) did not provide an implementation or evaluation of the method they proposed. In fact, their method requires a "mass" hyperparameter for each Linear, Embed, and Conv layer (see their section 4.1), so it isn't actually clear to us how one would implement this method without a lot of additional tuning or some other procedure for choosing these hyperparameters. Since the authors did not provide experiments themselves, we do not think it is necessary to compare against this method.
>
> On the LMO variants, did you have a specific method in mind? We have compared against Muon, Scion, PolarGrad, and many other variations, and we are not aware of any other well-established LMO methods.

---

### Official Review · Reviewer_VeqW · 2025-10-30

**Soundness:** 3
**Presentation:** 4
**Contribution:** 3
**Rating:** 6
**Confidence:** 3

**Summary:**

- Presents a unified view of Muon with Adam on a subset of parameters (MuonAdam) using non-Euclidean gradient descent, and places several existing opimizers, such as MuonAdam, Scion, and PolarGrad, within the same framework.
- Generalizes the model-based momentum method for adaptive learning rates (Momo) to arbitrary norms; and plug it into models on the non-Euclidean gradient descent framework.
- Systematically evaluates variants categorized by steepest-descent types (regularized or constrained), norm choices for Muon and Adam, finding that MuonMax-Momo achieves strong validation scores across a wider range of learning rates.

**Strengths:**

- Places MuonAdam and several other optimizers on the same non-Euclidean gradient descent framework.
- Combines Momo with the framework to achieve robustness to LR choices.
- Proposes MuonMax-Momo, which demonstrates increased LR-tuning robustness while keeping near-identical memory overhead and 5% additional wall-clock time per step compared with Muon.

**Weaknesses:**

- The paper unifies prior methods under the framework of non-Euclidean gradient-descent. Could this framework also help explain the theoretical properties of those methods? For example, the Momo paper includes a convergence analysis for Momo(-SGD) but not for Momo-Adam. How does this framework help understand the convergence properties for the algorithms such as MuonMax-Momo?
- Without theoretical analysis, evaluation of a paper may rely on empirical results. However, because the experiments were conducted only on GPT-2–based models and were limited to two datasets, it is difficult to confirm that the proposed approach will generalize broadly.

**Questions:**

Please refer to the weaknesses.

---

> ### Author Response · Authors · 2025-11-25
>
> Thank you for your helpful review. Below we have responded individually to your points.
>
> **Q1**: The paper unifies prior methods under the framework of non-Euclidean gradient-descent. Could this framework also help explain the theoretical properties of those methods? For example, the Momo paper includes a convergence analysis for Momo(-SGD) but not for Momo-Adam. How does this framework help understand the convergence properties for the algorithms such as MuonMax-Momo?
>
> **A1**: Thanks for the question. Because the RSD variants are essentially just gradient descent in a different geometry, the standard convergence analysis of (deterministic) gradient descent for both convex and non-convex functions will also apply to any RSD variant, under the assumption of smoothness of the objective function. You can see this, for example, in Section 3 of [1]. For stochastic variants or CSD variants, the classical proofs do not apply verbatim, but several works (Li & Hong 2025; Kovalev 2025, Riabinin et al 2025) have given convergence proofs for spectral descent in the stochastic setting, which we cited in our Related Work. Our framework might, in theory, be used to extend such convergence proofs for CSD on product spaces. We should also point out that, since MuonMax is based on a norm that changes over time, we would run into similar difficulties to prove convergence of MuonMax-Momo as the Momo paper may have for Momo-Adam. In summary, while our framework provides a means for analyzing steepest descent over ALL network parameters, this alone is not sufficient to prove convergence of algorithms like MuonAdam, due to difficulties with changing norms, stochasticity, and update normalization. If you find this discussion interesting, we can add it to the paper.
>
> **Q2**: Without theoretical analysis, evaluation of a paper may rely on empirical results. However, because the experiments were conducted only on GPT-2–based models and were limited to two datasets, it is difficult to confirm that the proposed approach will generalize broadly.
>
> **A2**: Thanks for the suggestion. We have expanded the scope of our experiments by adding new baselines, an MoE architecture for next token prediction, and image classification experiments with ResNets. Our results show that our proposed algorithms exhibit similar or improved performance compared to baselines while being significantly more robust to learning rate tuning. Please see the general response for a summary of our experiments.
>
> [1] Kelner, Jonathan A., et al. "An almost-linear-time algorithm for approximate max flow in undirected graphs, and its multicommodity generalizations." Proceedings of the twenty-fifth annual ACM-SIAM symposium on Discrete algorithms. Society for Industrial and Applied Mathematics, 2014.

---

### Official Review · Reviewer_jNaZ · 2025-10-30

**Soundness:** 3
**Presentation:** 3
**Contribution:** 4
**Rating:** 6
**Confidence:** 4

**Summary:**

This manuscript challenges gradient descent optimization techniques. Based on alternative norm computation, the authors propose MuonMax and Momo, which enable improved insensitivity to hyperparameter tuning, specifically on the learning rate. Experiments demonstrate these improvements.

**Strengths:**

- I appreciate the learning rate sweep results, such as Figure 1, which clearly demonstrates the proposed method succeeds in achieving robust hyperparameter tuning.
- Indeed, the hyperparameter tuning is an important issue, especially for large models. The learning rate is a core hyperparameter, and its insensitivity is expected to contribute to this field.
- Indeed, the transformers exhibit different properties for optimization. The authors performed targeted experiments on transformers, especially for LLM tasks.

**Weaknesses:**

- The experiments were performed for only two models and two datasets, which might be insufficient. The optimizers for comparison are limited to specific ones such as Muon and Scion.
- Although the results are convincing in robust hyperparameter tuning, the demonstration is focused only on the final validation loss. Is it possible to demonstrate other indices, such as practical ones? I think certain practitioners may want to capture the performance more practically, but the value of loss is difficult to understand on an absolute scale. It is also difficult to understand whether it corresponds to sufficient convergence or is still far from convergence.
- Experiments on more practical scenarios when learning rate tuning becomes difficult would enhance the results. For example, increasing mini-batch size affects tuning of learning rate; a sweep graph with mini-batch size, as well as learning rate, would make this manuscript further convincing.
- How about hyperparameter tuning sensitivity on \beta in the Adam family?
- Please check the following mathematics.
    - At Line 769, I think it should be $+r_t \Delta_t$, not $-r_t \Delta_t$.
    - At Line 938, it should be $r_t \Delta_t$, not $r_t + \Delta_t$.
    - For Line 8 of Algorithm 3, the last term should be g_t^\theta with \theta_t, not with m_t^\theta, to be compatible with Eqs. 85-88.
    - The sign of LMO is confusing. Specifically, when following the standard definition, LMO should be already negative, whose gradient descent becomes rather ascent. How about writing LMO at Eq. 68 to include a negative sign and writing the sign of LMO to be +, not -, at Eqs. 22 and 24? This choice depends on the authors but might improve readability.
- Writing should be improved.
    - “Furthermore” → “Furthermore, ” at Line 131.
    - “was” → “were” at Line 165.
    - “Schaipp et al. (2024)” should be written with \citep.
    - “the the final loss” → “the final loss” at Line 455.
    - “increases the lost” → “increases the loss” at Line 482.
    - “this is function is” → “this function is” at Line 1010.
    - For the caption of Figure 1, GPT2-Large should be 774M, not 124M.
- For the source code, the authors write HybridProductNorm, which computes lmo_dict. For this code, LMO is computed as a single constant for muon, whereas others are allowed to be layerwise constant. This requires explanation.

**Questions:**

Please see the weaknesses above. My score is based on the assumption that all typos are corrected in the revised manuscript.

---

> ### Author Response · Authors · 2025-11-25
>
> Thank you for your efforts in reviewing our paper. Below we have responded to your individual points.
>
> **Q1**: The experiments were performed for only two models and two datasets, which might be insufficient. The optimizers for comparison are limited to specific ones such as Muon and Scion.
>
> **A1**: Thank you for the suggestion. We have included additional experiments with four new baselines (AdamW, Lion, Adan, Sophia), a MoE architecture, and image classification with ResNets. Please see our general response for a summary of the new experiments.
>
> **Q2**: Although the results are convincing in robust hyperparameter tuning, the demonstration is focused only on the final validation loss. Is it possible to demonstrate other indices, such as practical ones? I think certain practitioners may want to capture the performance more practically, but the value of loss is difficult to understand on an absolute scale. It is also difficult to understand whether it corresponds to sufficient convergence or is still far from convergence.
>
> **A2**: To make the results more interpretable, we also included the best perplexity reached by each method in Tables 6, 7, 8. To judge whether models have converged, our original submission contains loss curves for the last 40% of training steps in Figure 1(b) and Figure 4. The loss curves suggest that the models are near convergence, though the loss may be made somewhat smaller by additional training. As part of our experiments with new baselines, we also included loss curves for FineWeb1B in Figure 6.
>
> **Q3**: Experiments on more practical scenarios when learning rate tuning becomes difficult would enhance the results. For example, increasing mini-batch size affects tuning of learning rate; a sweep graph with mini-batch size, as well as learning rate, would make this manuscript further convincing.
>
> **A3**: Thank you for the suggestion. We added additional experiments to evaluate our proposed algorithms and baselines with five different batch sizes, and we found the results to be consistent with our original experiments in the main paper: MuonAdam-Momo and MuonMax-Momo are more robust to LR tuning for all batch sizes, and for three out of five batch sizes these algorithms outperform the baselines for every learning rate we tried.
>
> **Q4**: How about hyperparameter tuning sensitivity on \beta in the Adam family?
>
> **A4**: Since the $\beta$ parameters of Adam are usually set to standard values, and to keep the cost of experiments reasonable, we use common settings of the $\beta$ parameters and focus on the sensitivity with respect to the learning rate(s). Also, Adam's sensitivity to the choice of the $\beta$ parameters was studied extensively in (Orvieto & Gower, 2025).
>
> **Q5**: Math and writing improvements.
>
> **A5**: Thank you for your thorough reading of our draft! We have corrected all typos and formatting issues. On the sign convention of the LMO definition, we agree that changing the definition would clean up a lot of negative signs, but we have decided to stay with the current definition to stay consistent with previous work like (Pethick et al, 2025).
>
> **Q6**: For the source code, the authors write HybridProductNorm, which computes lmo_dict. For this code, LMO is computed as a single constant for muon, whereas others are allowed to be layerwise constant. This requires explanation.
>
> **A6**: Thank you for taking the time to look at the source code. We do not entirely understand your question, but we can try to provide information about the code snippet you referenced. The HybridProductNorm object is used to compute the LMO and dual of the following norm: $$\lVert (v_1, \ldots, v_L, v_{L+1}) \rVert_{\text{hyb}} = \sqrt{\left( \max_{\ell \leq L} v_{\ell} \right)^2 + v_{L+1}^2},$$ which is the outer norm corresponding to $\lVert \cdot \rVert_{\text{MM}}$ (defined in Eq (17)), that is, the norm for which MuonMax is the steepest descent. The HybridProductNorm computes the LMO and dual in terms of the layer-wise LMOs and duals by implementing Lemma 3.3 exactly. The resulting algorithm is exactly MuonMax(-Momo) as we specified in Algorithm 3; there is no difference between the algorithm specified in our paper versus that implemented in the code.
>
> If we have not answered your question, please feel free to clarify. You said that LMO is computed as a single constant for muon, whereas others are allowed to be layerwise constant. Can you be more specific about which LMO you refer to here, perhaps the LMO of a single layer's gradient or the LMO of the outer product norm? Also, which "others" do you refer to? We are happy to continue the conversation until your question is answered, thanks.

---

### Official Review · Reviewer_kwnU · 2025-10-31

**Soundness:** 2
**Presentation:** 2
**Contribution:** 2
**Rating:** 2
**Confidence:** 3

**Summary:**

The paper develops a unified steepest-descent view of Muon-type optimizers by choosing per-layer norms, an across-layer product norm, and optional normalization. Within this framework it formalizes practical Muon+Adam (MuonAdam) as constrained steepest descent, extends model-truncation momentum (Momo) to arbitrary norms, and proposes MuonMax, a new regularized variant induced by a max-spectral or ada2 product norm. Language-modeling experiments on GPT-2 scales suggest that Momo-augmented variants, especially MuonMax-Momo, achieve similar or better final loss with wider learning-rate basins, while a stale dual-norm trick preserves accuracy with modest overhead.

**Strengths:**

1. Clear theoretical unification. The paper characterizes MuonAdam as constrained steepest descent under a specific product norm, aligning spectral-norm LMOs for matrices with Adam-style updates for non-matrix parameters in a single principled step and opening a coherent design space.

2. Practical and robust variant with concrete derivations. MuonMax is explicitly defined, its Momo version admits a closed-form update, and experiments show broad learning-rate robustness and reduced seed variance relative to strong baselines.

**Weaknesses:**

1. Efficiency overhead is only partially quantified. The stale nuclear-norm approximation reduces MuonMax-Momo overhead from roughly eleven percent to about five percent per step versus MuonAdam, but the paper does not report end-to-end throughput or time-to-target loss across hardware and precision regimes, and the reliance on PolarExpress in bfloat16 introduces possible stability and accuracy tradeoffs that are not analyzed.

2. Scope limited to autoregressive language model pretraining. All experiments are on GPT-2-style models and two text corpora, with no tests on architectures or modalities where parameter partitions and curvature differ, such as ViTs, ConvNets, MoEs, or sequence-to-sequence models. The paper cites closely related modular-norm methods but does not include a direct baseline from that family.

3. Limited baseline coverage under matched tuning budgets. Comparisons emphasize Muon family variants, but the paper does not present compute-normalized results against strong first-order baselines such as AdamW, Lion, Adan, or Sophia with matched hyperparameter search budgets. Because the paper highlights “wide learning-rate basins” as a key advantage, the absence of equal-budget LR and scheduler sweeps for these baselines weakens the central empirical claim.

**Questions:**

1. Will you add compute-normalized, equal-budget comparisons against strong first-order baselines such as AdamW, Lion, Adan, and Sophia, reporting identical hyperparameter search grids, scheduler options, and seeds, and measuring both final loss and time-to-target?

2. Since wide learning-rate basins are a central claim, can you quantify basin width uniformly across methods using the same grid and seed protocol, and report success rates, median best LR, and sensitivity surfaces for matrix and non-matrix step sizes?

---

> ### Author Response · Authors · 2025-11-25
>
> Dear Reviewer,
> We thank you for your suggestions, and clear review. We also find your request for specific additional experiments to be reasonable, and we have been working hard to produce several new experimental results.
>
> **Q1**: Efficiency overhead is only partially quantified. The stale nuclear-norm approximation reduces MuonMax-Momo overhead from roughly eleven percent to about five percent per step versus MuonAdam, but the paper does not report end-to-end throughput or time-to-target loss across hardware and precision regimes, and the reliance on PolarExpress in bfloat16 introduces possible stability and accuracy tradeoffs that are not analyzed.
>
> **A1**: Thank you for your suggestions. For the throughput and time to target loss metrics, see our A4. For the points about precision, we point out that the original Muon implementation also uses bfloat16 to compute the polar factor with polynomial methods (Jordan et al, 2024b), as does Scion (Pethick et al, 2025), and indeed every other Muon implementation we are aware of. Therefore, the stability and accuracy tradeoffs in terms of precision are not something introduced by our work, and while this is an important practical topic for Muon in general, we view this as orthogonal to the present work.
>
> **Q2**: Scope limited to autoregressive language model pretraining. All experiments are on GPT-2-style models and two text corpora, with no tests on architectures or modalities where parameter partitions and curvature differ, such as ViTs, ConvNets, MoEs, or sequence-to-sequence models. The paper cites closely related modular-norm methods but does not include a direct baseline from that family.
>
> **A2**: We have expanded the scope of our experiments by including an MoE architecture (Qwen2Moe) for language modeling, and two ResNets for image classifications. Please see our general response for a summary of the new experiments.
>
> As for the modular norm methods, we reiterate out statement from the related work that we are not aware of any implementation or experimental evaluation of steepest descent with respect to the modular norm paper. (Bernstein & Newhouse, 2024b) proposed this algorithm, but does not actually contain any experiments, and the algorithm they propose involves many additional hyperparameters, e.g. the "mass" hyperparameters for each Linear, Embed, or Conv layer from their Section 4.1. It is not clear to us how one should implement this algorithm or set these additional hyperparameters.
>
> **Q3**: Limited baseline coverage under matched tuning budgets. Comparisons emphasize Muon family variants, but the paper does not present compute-normalized results against strong first-order baselines such as AdamW, Lion, Adan, or Sophia with matched hyperparameter search budgets. Because the paper highlights “wide learning-rate basins” as a key advantage, the absence of equal-budget LR and scheduler sweeps for these baselines weakens the central empirical claim.
>
> **A3**: Thanks for the suggestions. We agree that the experiments will be improved with these additional baselines and metrics. Please see our A4 for a discussion of our new experiments with additional baselines, and our A5 for a discussion of our quantification of LR basin width.
>
> **Q4**: Will you add compute-normalized, equal-budget comparisons against strong first-order baselines such as AdamW, Lion, Adan, and Sophia, reporting identical hyperparameter search grids, scheduler options, and seeds, and measuring both final loss and time-to-target?
>
> **A4**: Thank you for the very concrete suggestions for improving our experiments. We have added new experiments with the four baselines you suggested, and each baseline was tuned with the same grid, same seeds, same scheduler, and with equal or greater compute budget than our proposed methods. We measured final loss, and to further quantify efficiency we measured time to target loss, tokens to target loss, time per training step, and end-to-end token throughput. These additional metrics are now included in Tables 6, 7, 8 of Appendix D. Please see our general response for a summary of these experiments, and let us know if we have resolved your concerns.

---

> ### Author Response · Authors · 2025-11-25
>
> **Q5**: Since wide learning-rate basins are a central claim, can you quantify basin width uniformly across methods using the same grid and seed protocol, and report success rates, median best LR, and sensitivity surfaces for matrix and non-matrix step sizes?
>
> **A5**: Thank you, quantifying basin width was a good idea and we believe this has improved our experimental results. The basin width of every method for our language modeling experiments is included in Tables 6, 7, 8, and the results conclusively show that our MuonAdam-Momo and MuonMax-Momo exhibit much wider loss sensitivity curves: for FineWeb1B/GPT2-Small, the basin widths for these two algorithms is 2.5 and 3.5, respectively, compared to 1.5 for MuonAdam and 1 for AdamW. For SlimPajama1B/GPT2-Large, the gap is even bigger. We hope that this quantitative evidence convinces you of the validity of our claims of robustness. Please let us know if we have resolved your concerns.

---

### Official Review · Reviewer_xjd6 · 2025-11-01

**Soundness:** 3
**Presentation:** 3
**Contribution:** 2
**Rating:** 4
**Confidence:** 4

**Summary:**

This paper presents a unified theoretical framework for understanding the Muon optimizer and its variants as forms of steepest descent. The framework is characterized by three main design choices: the norm for each parameter group, the product norm used to aggregate norms across groups, and the type of steepest descent (constrained vs. regularized). Based on the framework, the authors propose a new variant, MuonMax. Furthermore, the paper generalizes an adaptive step-size method based on model truncation (Momo) to work with arbitrary norms. Experiemnts are conducted on GPT-2 models.

**Strengths:**

1. The main strength of the paper is the development of a clear and unifying theoretical framework. Casting Muon-style optimizers as non-Euclidean steepest descent with choices of product norms and descent types is an insightful contribution.
2. The technical contribution of extending Momo to operate with arbitrary norms is potentially useful for applying model truncation techniques to a wider range of optimizers.

**Weaknesses:**

1. While the framework is theoretically elegant, it defines a large design space (norm for each parameter group, product norm, CSD vs. RSD) without providing strong principles or theoretical intuition for navigating it. The choice of MuonMax's design feels arbitrary rather than being a principled consequence of the theory.
2. The proposed MuonMax optimizer is not sufficiently motivated. It is presented as one alternative in the design space, but the paper offers no theoretical argument for its superiority. Empirically, its performance is inconsistent. For instance, vanilla MuonMax is outperformed by MuonAdam as shown in Fig. 5. The paper's central claims heavily rely on its combination with Momo, suggesting MuonMax itself may not be a robust improvement.
3. The paper's solution to learning rate sensitivity, Momo, introduces its own hyperparameters. Fig. 3 shows that the final performance of both MuonAdam-Momo and MuonMax-Momo is sensitive to the choice of $F_*$. This undermines the claim of reducing tuning complexity, as it appears to merely shift the burden from tuning $\eta$ to tuning $F_*$.
4. The experimental evaluation is narrow. The comparison is entirely focused on Muon variants within the proposed framework, missing widely-used baselines like AdamW. Furthermore, using only GPT-2 architectures limits the claims of generality.

**Questions:**

See above.

---

> ### Author Response · Authors · 2025-11-25
>
> Thank you for your helpful comments. Below we have addressed your points individually.
>
> **Q1**: While the framework is theoretically elegant, it defines a large design space (norm for each parameter group, product norm, CSD vs. RSD) without providing strong principles or theoretical intuition for navigating it. The choice of MuonMax's design feels arbitrary rather than being a principled consequence of the theory.
>
> **A1**: We agree, the theory does not provide a reason why MuonMax should be any better or worse than Muon, or any other variant. This is a major open problem of this line of research which has been motivated by the success of Muon: why prefer one norm for steepest descent over another when training deep networks? We do not claim to be handling this major question. Instead, our aim here was to show that there are more design choices and practical variants of Muon than are currently not being considered (unconstrained version such as MuonMax), how to implement these variants efficiently (stale computations), how to use truncation with these methods, and present an empirical evaluation for the algorithms in this design space.
>
> The only fact we can point to motivating MuonMax, it uses the most theoretically ground product norm for the hidden weight matrices, which is the max over the layers (Bernstein & Newhouse, 2024b), and a practical product norm for the other matrices, which is the adaptive $\ell_2$ norm corresponding to Adam. In this sense, it is not so surprising it is one of the most robust variants.
>
> **Q2**: The proposed MuonMax optimizer is not sufficiently motivated. It is presented as one alternative in the design space, but the paper offers no theoretical argument for its superiority. Empirically, its performance is inconsistent. For instance, vanilla MuonMax is outperformed by MuonAdam as shown in Fig. 5. The paper's central claims heavily rely on its combination with Momo, suggesting MuonMax itself may not be a robust improvement.
>
> **A2**: We again agree that MuonMax without Momo is not superior to the baselines, and we did not mean to give this impression. In the paper, we only emphasize that the design space is bigger than what is being considered, and argue for the superior performance of MuonAdam-Momo and MuonMax-Momo compared to baselines, as these two algorithms significantly improve the learning rate sensitivity and sometimes final loss.
>
> **Q3**: The paper's solution to learning rate sensitivity, Momo, introduces its own hyperparameters. Fig. 3 shows that the final performance of both MuonAdam-Momo and MuonMax-Momo is sensitive to the choice of $F_*$. This undermines the claim of reducing tuning complexity, as it appears to merely shift the burden from tuning $\eta$ to tuning $F_*$.
>
> **A3**: We agree that $F_*$ is an additional hyperparameter, but its tuning is immensely easier than that of the learning rate, first since there is always a natural, safe guess of $F_* = 0$, and second because the algorithm performance is actually quite insensitive to the value of $F_*$. The left side of Fig 3 shows that, for all learning rates less than $1.0$, the loss curves for all $F_*$ between $0$ and $3.2$ are actually overlapping! There is essentially no change in the performance of MuonAdam-Momo even as $F_*$ varies from the safe guess of $0$ to the target loss of $3.2$. For the right side of Fig 3, we see that MuonMax-Momo is slightly more sensitive to the choice of $F_*$, but the sensitivity with respect to $F_*$ is still much lower than that with respect to $\eta$: when $\eta = 0.01$, the loss only changes from $3.58$ to $3.61$ as $F_*$ varies over the entire grid from $0$ to $3.2$. This variation is smaller than the change in loss when $\eta$ is moved a single position in the grid.
>
> Figure 3 shows that even without tuning $F_*$ at all, the safe choice $F_* = 0$ always performs close to optimal. Furthermore, in any large scale experiment, one always uses a scaling law. Scaling laws provide a target loss value that will be reached for the given model size and number of tokens. Thus in practice there is no need to tune $F_*$ at all, either by using a scaling law, or simply setting to $F_*=0.$
>
> **Q4**: The experimental evaluation is narrow. The comparison is entirely focused on Muon variants within the proposed framework, missing widely-used baselines like AdamW. Furthermore, using only GPT-2 architectures limits the claims of generality.
>
> **A4**: Thank you for the suggestion. We have added more experiments including more baselines (AdamW, Lion, Adan, Sophia), another architecture for language modeling (Qwen2-MoE) and image classification experiments with ResNet. We believe that our new results show that our proposed algorithms outperform strong baselines in a diverse range of settings. Please see our general response for a summary of the new experiments.

---

### Author Response · Authors · 2025-11-25
**Response to All Reviewers**

Thank you to all of the reviewers for your effort and helpful feedback. We apologize for the delay in responding, we have been working hard including. your suggestions, and have now added several new experiments to the paper, which we summarize below:
- Additional baselines for FineWeb1B: In response to reviewer kwNU, we added to our GPT2-Small/FineWeb1B setting evaluations for AdamW, Lion, Adan, and Sophia. We ensured that these additional baselines were given as much or more computational budget for tuning hyperparameters than our proposed algorithms, and we found that only AdamW is competitive with the Muon-type methods, though even AdamW is outperformed by a reasonable margin by these Muon-type methods. See Section D.1.1 for the full results.
- Additional metrics: In response to reviewer kwNU, for FineWeb1B, SlimPajama1B, and SlimPajama6B, we further quantified both the efficiency and the learning rate robustness of each algorithm. Efficiency is reported in terms of token throughput, time per step, time to reach a target loss, and tokens to reach a target loss. We found that the range of competitive learning rates for our proposed algorithms is multiple orders of magnitude larger than that of baselines. See Tables 5-7 in Appendix D.1.2 for the full results and discussion.
- Additional architecture for FineWeb1B: In response to all five reviewers, we additionally trained two variants of a Qwen2-MoE architecture on FineWeb1B. We found that MuonAdam-Momo and MuonMax-Momo with tuned learning rates achieve smaller loss than Muon and Scion, but MuonAdam-Momo and MuonMax-Momo were again more robust to the choice of learning rates, see Section D.1.4 for the full results.
- Evaluation of different batch sizes: In response to reviewer jNaZ, we evaluated our proposed algorithms and main baselines with five different batch sizes, and found the results to be consistent with those of our original experiments: our proposed algorithms achieve lower losses and greater robustness to the choice of learning rate than baselines.
- Image classification experiments: In response to all five reviewers, we ran experiments with a ResNet20 on CIFAR-10 and a ResNet110 on CIFAR-100, comparing SGD with momentum and AdamW against Muon, Scion, MuonAdam-Momo, and MuonMax-Momo. We found that MuonMax-Momo achieves the lowest loss and highest validation accuracy for both settings, and for ResNet20/CIFAR-10 MuonMax-Momo achieves the smallest loss for every choice of learning rate we tried. Full results in Appendix D.3.

These additions are shown in our submission as red text. Thank you again for your work, and please let us know if we have answered your questions.

---

### Meta-Review · Area_Chair_wg4t · 2026-01-08

**Summary:**

This paper received 5 reviews. The reviewers (score/confidence) are: `xjd6 (4/4), jNaZ (6/4), kwnU (2/3), VeqW (6/3), 27dR (6/3)`. Their major concerns:

Methodology:

- The theoretical framework lacks principled guidelines for navigating the "large design space", and the design of the proposed MuonMax is arbitrary rather than theoretically motivated `xjd6 (4/4)`.

- No theoretical convergence analysis is provided for MuonMax-Momo or Momo-Adam, and the stale dual-norm approximation lacks "stability/error bounds" `VeqW (6/3) 27dR (6/3)`.

Experiments:

- The experimental scope is narrow, limited to GPT-2 architectures and two datasets without testing on ViTs, ConvNets, or larger-scale LLMs (all the reviewers questioned this: `xjd6 (4/4) kwnU (2/3) VeqW (6/3) 27dR (6/3)`).

- Mainstream baselines such as AdamW, Lion, Adan are missing, and compute-normalized comparisons with equal hyperparameter search budgets are not provided `xjd6 (4/4) kwnU (2/3)`.

- Evaluation metrics are limited to validation loss, lacking practical indicators (e.g., time-to-target loss, downstream task performance) `jNaZ (6/4) 27dR (6/3)`.

- End-to-end throughput and hardware/precision regime analysis are not quantified `kwnU (2/3)`.

**Reviewer Concerns:**

The two negative reviewers:

- xjd6 (4/4): The reviewer poses a few critical concerns regarding the method, hyper-params, and limited empirical results (only comparing with the baseline, not including AdamW etc optimizers that are widely used counterpart methods). The authors added the new results during the rebuttal and made the clarifications. I checked the discussions and feel the rebuttal is not very strong. The authors also did not discuss how the proposed method performs in the newly added experiments. Overall, I believe the concerns of this reviewer are still outstanding.

- kwnU (2/3): The reviewer questioned the efficiency overhead of the method and requested end-to-end throughput. The authors added the throughput in rebuttal. Also, the authors added new results on ResNets and other optimizers. These should also address many of the concerns. In general, I believe this reviewer's concerns are mostly addressed, but not to a very strong extent -- e.g., the reviewer requested  results on ViTs but this is not met during rebuttal.

**Reviewer Scores:**

The two negative reviewers:

- xjd6 (4/4): The concerns are still outstanding. Therefore, the score is probably maintained.

- kwnU (2/3): Most of the concerns are addressed. This reviewer may raise the score to 4 or 6. Raising to 6 is not very likely, considering the rebuttal is not very strong.

---

### Decision · Program_Chairs · 2026-01-26

Reject